# qTAG: an adaptable plasmid scaffold for CRISPR-based endogenous tagging

Reuben Philip [1,2,4], Amit Sharma [1,4], Laura Matellan [1], Anna C Erpf [1], Wen-Hsin Hsu [1], Johnny M Tkach [1], Haley D M Wyatt [3] & Laurence Pelletier [1,2✉]

## Abstract

**Endogenous tagging enables the study of proteins within their native regulatory context, typically using CRISPR to insert tag sequences directly into the gene sequence. Here, we introduce qTAG, a collection of repair cassettes that makes endogenous tagging more accessible. The cassettes support N- and C-terminal tagging with commonly used selectable markers and feature restriction sites for easy modification. Lox sites also enable the removal of the marker gene after successful integration. We demonstrate the utility of qTAG with a range of diverse tags for applications in fluorescence imaging, proximity labeling, epitope tagging, and targeted protein degradation. The system includes novel tags like mStayGold, offering enhanced brightness and photostability for live-cell imaging of native protein dynamics. Additionally, we explore alternative cassette designs for conditional expression tagging, selectable knockout tagging, and safe-harbor expression. The plasmid collection is available through Addgene, featuring ready-to-use constructs for common subcellular markers and tagging cassettes to target genes of interest. The qTAG system will serve as an open resource for researchers to adapt and tailor their own experiments.**

**Keywords** CRISPR; Endogenous Tagging; Gene-editing
**Subject Categories** DNA Replication, Recombination & Repair; Methods & Resources

## Introduction

Gene overexpression has long been a crucial tool in molecular biology, offering insight into gene function and how genes contribute to disease. Nevertheless, this approach has its constraints. Excessive protein abundance can result in mislocalization, causing interactions to deviate from their native state and ultimately manifest in ectopic phenotypes (Bolognesi et al, 2016; Farhan et al, 2008; Makanae et al, 2013; Tang and Amon, 2013;

Vavouri et al, 2009). This emphasizes the importance of examining proteins at their native expression levels, where a more accurate comprehension of a protein's localization, dynamics, and interactions can be studied. Since its discovery in 2012, the advent of clustered regularly interspaced short palindromic repeats (CRISPR) alongside its programmable Cas endonucleases has sparked a transformative revolution in molecular genetics (Jinek et al, 2012). Using a single guide RNA (sgRNA), this technology enables the editing of DNA by directing the Cas endonucleases to a specific genomic locus and allowing for targeted edits to be made. This is especially relevant in the context of endogenous tagging, where the insertion of large transgenes encoding functional tags into the reading frame of a gene of interest (GOI) facilitates the study of proteins, without the potential consequences of overexpression.

Cas endonucleases, such as Cas9, exhibit a high level of precision in identifying DNA sequences to cleave. When guided by the sgRNA, Cas9 locates a DNA sequence matching the targeting sequence adjacent to a protospacer adjacent motif (PAM). This prompts Cas9 to initiate a sequence-specific double-strand break (DSB), after which the cell's innate DNA repair mechanisms are triggered to engage in repair (Paques and Haber, 1999; Urnov et al, 2010). In the absence of a template to guide the repair, the DSB can be resolved by joining the two ends together through nonhomologous end-joining (NHEJ) (Bibikova et al, 2002; Chang et al, 2017; Moore and Haber, 1996). This mechanism frequently incorporates base pair (bp) insertions or deletions (indels), that can disrupt the gene's coding region and as such has been leveraged to effectively disrupt gene function (Hart et al, 2015). Two alternative DSB repair pathways routinely exploited for precise gene insertions are: homology-directed repair (HDR) and microhomology-mediated end joining (MMEJ). Insertions using HDR require a donor DNA template containing long stretches of homologous sequence flanking the intended modification site to mediate repair (Jasin and Rothstein, 2013). In contrast, MMEJ-targeted gene editing relies on the binding of microhomologies of ~5–25 bp of complementary sequence to engage repair (Sakuma et al, 2016; Sfeir and Symington, 2015). Targeting these pathways improves editing accuracy over NHEJ, albeit at the cost of diminished efficiency, since these pathways exhibit lower activity levels and are only activated during specific stages of the cell cycle (Taleei and Nikjoo 2013). From an execution standpoint, they also necessitate the

[1]Lunenfeld-Tanenbaum Research Institute, Mount Sinai Hospital, Toronto, ON M5G 1X5, Canada. [2]Department of Molecular Genetics, University of Toronto, Toronto, ON M5S 3E1, Canada. [3]Department of Biochemistry, University of Toronto, Toronto, ON M5S 1A8, Canada. [4]These authors contributed equally: Reuben Philip, Amit Sharma. ✉E-mail: pelletier@lunenfeld.ca

supply of loci-specific homologous sequences, which can be difficult to amplify and integrate into repair constructs. Collectively, when attempting to insert large transgenes, these limitations often result in extremely low integration rates, making the isolation of homozygous cell lines both challenging and time-consuming (Artegiani et al, 2020; Fueller et al, 2020; He et al, 2016; Roberts et al, 2017).

Instead of enhancing CRISPR or delivery components directly to address low integration rates, alternative approaches have long explored optimizing the donor construct itself to improve editing outcomes. This typically involves partitioning the donor cassette to include the desired tag or modification, along with a fluorescent or mammalian selectable marker to allow for easy enrichment of edited cells. Several methods have explored this concept (Artegiani et al, 2020; Fueller et al, 2020; Lin et al, 2019; Nabet et al, 2018; Nakamae et al, 2017; Perez-Leal et al, 2021; Sakuma et al, 2016; Supharattanasitthi et al, 2019), with donor designs being largely summarized into two groups: promoter-based and multicistronic repair cassettes. Promoter-based cassettes involve the insertion of a tag within the reading frame of a GOI followed by a distinct promoter sequence that drives the expression of the selectable marker. In contrast, multicistronic approaches make use of sequence elements like internal ribosome entry sites (IRES) (Jang et al, 1988; Pelletier and Sonenberg, 1988) or 2A "self-cleaving" peptide sequences (Kim et al, 2011; Luke et al, 2008) to facilitate efficient co-expression of the selectable marker.

Here, we've developed a set of optimized repair cassettes aimed at simplifying the process of editing and selecting endogenously tagged cell lines. Our "quickTAG" (qTAG) system provides a plasmid scaffold that can be implemented with straightforward cloning approaches and can be easily adapted to accommodate alternative tags and selection markers. This study details the design and utility of these plasmids, showcasing their ability to facilitate both fluorescent and non-fluorescent tagging. Additionally, we provide insights into strategies for efficiently generating homozygous cell lines as well as multiple rounds of editing within the same cell line. Finally, we demonstrate the versatility of these cassettes through alternative editing scenarios, expanding their applications beyond endogenous tagging to include conditional expression tagging, selectable knock-out tagging, and safe-harbor targeted expression. By providing convenient access to the qTAG library of plasmids through Addgene (https://www.addgene.org/Laurence_Pelletier/), our goal is to streamline the adoption of endogenous tagging and promote the adaptation of this plasmid library for future editing strategies and applications.

# Results

## A versatile scaffold to streamline endogenous tagging

The qTAG cassettes are built upon the core 2A-based cassette structure described previously (Nabet et al, 2018; Sakuma et al, 2016). The initial cassettes were designed and synthesized as gene fragments with the 2A sequence situated between a tag and a resistance gene. Due to its transcriptional linkage and short sequence length, the 2A-based approach was adopted over an IRES element or a promoter to co-express a selectable marker that would enrich edited cells. Four core features were integrated into

the cassette structure to ensure ease of use while maintaining adaptability (Fig. 1). These included: (1) Cassette designs specific to both the N- and C-terminus encoding regions, for tagging at either end of a gene. (2) To simplify cloning, specific cloning sequences were placed around the cassette. These sequences permit a uniform cloning approach regardless of the tag, resistance, or specific sequences associated with the GOI. These sequences were also designed to accommodate both long and short homology arms, targeting the HDR or MMEJ pathways for precise homology-based repair. (3) To enhance cassette adaptability, unique restriction sites were placed next to each critical sequence element within the cassette. This design allows for easy alteration through restriction cloning and permits future customization. (4) Finally, lox sites flank the selectable marker gene to enable the removal of the marker after transient introduction of the Cre recombinase (Albert et al, 1995; Sauer and Henderson, 1988). The mutant lox sites lox66 and lox71 were chosen to ensure irreversible deletion of the selectable marker gene and recovery of selection sensitivity (Zhang and Lutz, 2002). This is particularly useful in experimental conditions where only a single selectable marker can be used to sequentially tag genes.

The standardized cassette layout includes a selection of state-of-the-art tags to support a diverse array of assays. A panel of the brightest currently known monomeric fluorescent proteins covering the green, red, and far-red emission spectra enables observation and tracking of a target protein's endogenous localization through live or fixed cell imaging. These include mStayGold (Ando et al, 2024), mNeon (Shaner et al, 2013), moxGFP (Costantini et al, 2015), super-TagRFP (Mo et al, 2020), mScarlet (Bindels et al, 2017), and miRFP670nano3 (Oliinyk et al, 2022). The qTAG cassettes also support a wide variety of non-fluorescent tags. Proximity-dependent biotinylation enzymes, such as miniTurbo and ultraID, can be used to map the protein interactions that occur within close proximity of the tagged target protein (Branon et al, 2018; Kubitz et al, 2022). For targeted-degradation studies, dTAG can be employed to induce rapid and reversible protein depletion with high specificity (Nabet et al, 2018). Finally, epitope tagging offers a solution in situations where antibodies against a protein of interest are unavailable, ineffective, or cross-react with paralogs. For this, we generated tagging cassettes containing 3xFLAG, 3xHA, and V5 epitopes, for which high-quality detection reagents are available.

## Editing and enrichment of fluorescent knock-ins with qTAG cassettes

To assess the functionality of the cassette design and the effectiveness of the editing and enrichment strategy, we initially chose to tag the histone protein H2BC11 with the fluorescent protein moxGFP (Fig. 2). The C-terminal encoding region of H2BC11, was targeted using a sgRNA to permit the integration of a moxGFP-Puro cassette prior to the stop codon (Fig. 2A). We constructed moxGFP-Puro qTAG repair constructs containing both long and short homology arms in order to target the HDR and MMEJ pathways for insertion and enrichment in HEK293T cells (Fig. 2B). When surveying cells for green fluorescence as an indication of cassette integration, a small proportion of cells in the initial pool expressing the H2B-moxGFP fusion was observed when tagged using either cassette (Fig. 2C,D). After selection with puromycin, a strong enrichment was observed in populations

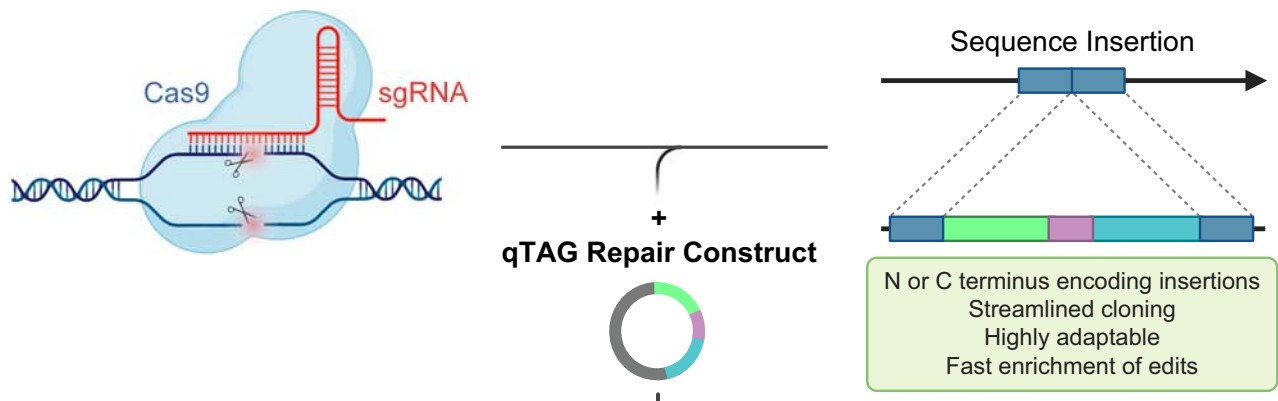

**qTAG Repair Construct**

**Sequence Insertion**

N or C terminus encoding insertions
Streamlined cloning
Highly adaptable
Fast enrichment of edits

## ① N and C Terminus Encoding Insertions

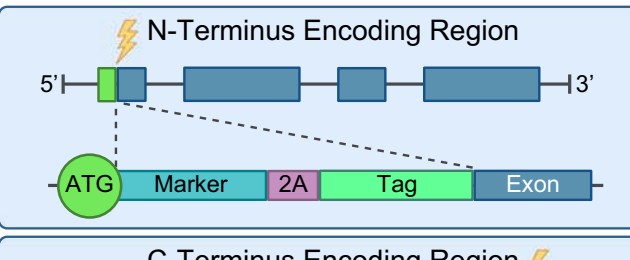

N-Terminus Encoding Region

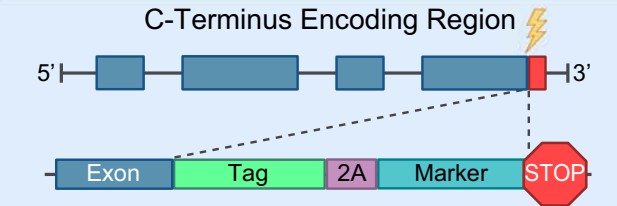

C-Terminus Encoding Region

## ② Easy Cloning for Homology Based Insertion

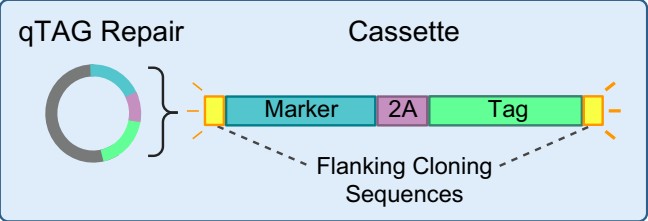

qTAG Repair Cassette

Flanking Cloning Sequences

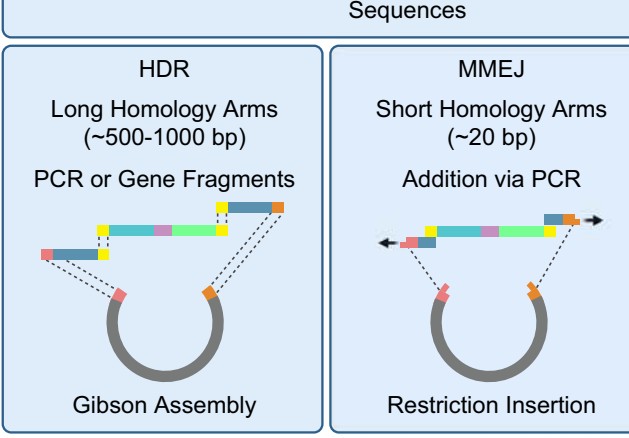

**HDR**

Long Homology Arms (~500–1000 bp)

PCR or Gene Fragments

Gibson Assembly

**MMEJ**

Short Homology Arms (~20 bp)

Addition via PCR

Restriction Insertion

## ③ Adaptable Cassette Design

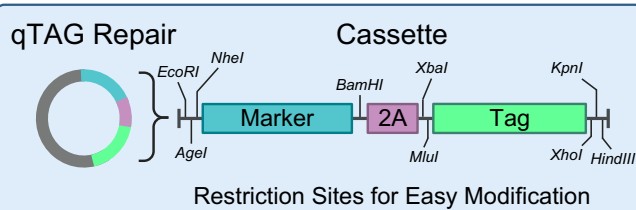

qTAG Repair Cassette

Restriction Sites for Easy Modification

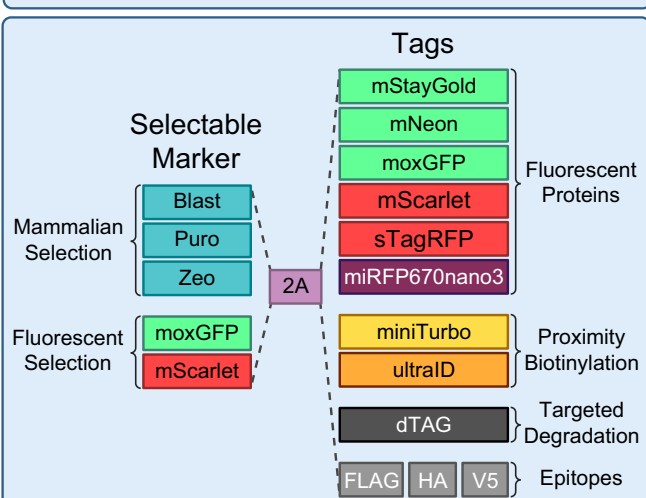

Tags

Selectable Marker

Mammalian Selection: Blast, Puro, Zeo

Fluorescent Selection: moxGFP, mScarlet

Fluorescent Proteins: mStayGold, mNeon, moxGFP, mScarlet, sTagRFP, miRFP670nano3

Proximity Biotinylation: miniTurbo, ultraID

Targeted Degradation: dTAG

Epitopes: FLAG, HA, V5

## ④ Selectable Marker Deletion

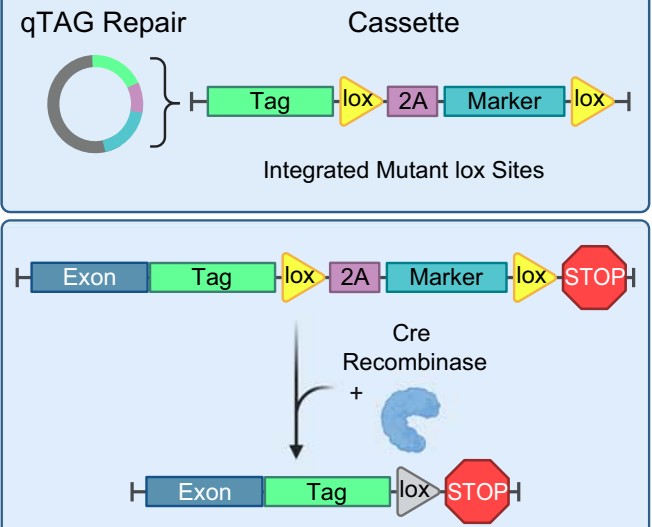

qTAG Repair Cassette

Integrated Mutant lox Sites

Cre Recombinase

**Figure 1. Features of qTAG cassettes.**

Four key features of the qTAG cassettes are depicted here. (1) 5′ and 3′ designs for N- and C-terminal modification; (2) the incorporation of uniform cloning sequences to streamline the cloning of long or short homology arms for HDR or MMEJ-targeted repair; (3) the inclusion of restriction sites near each genetic element in the cassette to conveniently swap tags or selectable markers; (4) the addition of lox sites on either side of the selectable marker to enable the marker gene removal.

edited via cassettes targeting HDR and MMEJ, as indicated by the shift in the distribution of cells displaying green fluorescence. We validated the target integration using fluorescence imaging to observe the localization of the endogenous fusion protein. Nuclear localization of H2BC11-moxGFP edited cells was confirmed through counterstaining with DAPI (Fig. 2E). Additionally, the presence of tagged alleles was verified by PCR using primers that anneal outside the homology arms. We observed a band shift consistent with the increased size of the qTAG-moxGFP-Puro cassette compared to the untagged allele (Fig. 2F). This tagging methodology was tested across various human cell lines, including HAP1, ARPE-19, and U-2OS cells (Fig. EV1a–d). Despite differing initial tagging rates, consistent enrichment of edited alleles via cassettes targeting the HDR and MMEJ pathway was observed post-selection with puromycin. Alternative selection markers using blasticidin and zeocin also demonstrated their equivalence in enriching cells with tagged alleles (Fig. EV1e,f). We also assessed the ability to introduce these insertions into H9 human embryonic stem cells. Repair constructs were devised to insert a qTAG-Blast-mScarlet cassette into the N-terminus encoding region of the tubulin gene TUBA1B. For editing, electroporation with precomplexed RNPs containing purified SpCas9 and an in-vitro transcribed sgRNA was used for delivery (Fig. EV1g). Following delivery, selection enrichment, and clonal expansion, mScarlet-TUBA1B expressing H9 cells were observed while still maintaining their pluripotent state, which was confirmed via staining against OCT4 (Fig. EV1h).

To gauge the efficiency of this cassette system, we aimed to examine the rates of accurate localization within selected pools, determine the allelic editing outcomes, and evaluate the final clonal results of these edited alleles within cells. For this, a panel of genes were tagged in RPE-1 PAC (−) cells using HDR-targeting cassettes containing the fluorescent protein mStayGold (Fig. EV2a). To assess the localization of these endogenous fusions, cells were quantified to determine the rates of either no signal localization, diffuse signal localization, or if the enriched cells had produced localized protein signals (Fig. EV2b). Over 80% of the populations across our fluorescently edited cell lines displayed localized fluorescent signals, with the tagging of ACTB resulting in a proportion of the cells that contained either no signal or mis-localized/diffuse signal within the final selection enriched population. These selected pools were also subjected to junction PCR, followed by amplicon sequencing to assess editing outcomes at the sequence level (Fig. EV2c). Amplicons at the sites with the expected insertions consistently showed precise editing, while those without insertions often displayed unexpected genetic alterations, likely repair outcomes triggered by the sgRNA through alternative DNA repair mechanisms. From a clonal perspective, there was considerable variability in allelic outcomes across this panel of edited cell lines. Genes like ACTB and PXN achieved approximately 30% homozygosity, whereas CLTC tagging showed no detectable homozygous cells (Fig. EV2d–g). These efficiency experiments were also conducted to compare editing outcomes between the use of HDR or

MMEJ targeting qTAG cassettes (Fig. EV3). Here, the H2BC11 gene was targeted for editing using a mStayGold-Puro cassette constructed with either long or short homology arms. Quantification of localization revealed a slight difference in localization rates between the MMEJ and HDR-targeted strategies (Fig. EV3a,b). Amplicon sequencing revealed that the MMEJ-targeted population had a higher proportion of mutated sequences in both the expected inserted and non-inserted amplicons, indicating an expected cost in editing precision compared to targeting the HDR pathway, for the convenience of easier homology arm construction (Fig. EV3c). The clonal genomic PCR confirmed the presence of more editing byproducts in the form of additional dominant bands beyond the two expected outcomes, suggesting additional insertions beyond the intended targets (Fig. EV3d,e).

## Selectable marker gene excision and serial gene tagging

Next, we sought to demonstrate the ability to recover antibiotic sensitivity and subsequently use a single mammalian resistance gene for serial gene-tagging. To do this, we targeted two genes: the tubulin protein TUBB4B with a C-terminus encoding insertion of a mNeon-Blast cassette and the histone protein H3C2 with a C-terminus encoding insertion of a miRFP670nano3-Blast cassette (Fig. 3A). Editing commenced with TUBB4B in ARPE-19 cells. After blasticidin selection, the pool was transiently transfected with a plasmid encoding Cre-2A-Puro to excise the blasticidin resistance gene. Following a brief puromycin selection for two days to enrich transiently transfected cells, an enriched pool of TUBB4B-Neon cells expressing the Cre recombinase was generated. This pool was subsequently seeded for clonal expansion, and a clonal homozygous cell line with a rescued sensitivity to blasticidin was isolated (Fig. 3B). PCR across the insertion junctions from cell lines representing the various phases of cell-line editing confirmed the presence of tagged alleles at the TUBB4B locus in the initial tagged pool (Fig. 3C). A molecular weight band shift consistent with the loss of the blasticidin sequence was noticed in the Cre-transfected pool as well as the clonal cell line suggesting successful excision of the blasticidin gene. To further verify that the reading frame of TUBB4B was not altered, the TUBB4B locus was amplified from the TUBB4B-mNeon clonal cell line and Sanger sequenced. Due to the use of mutant lox sites, if the sequence between these elements was excised correctly, a mutant low-binding lox71/66 site would form. We amplified and sequenced the TUBB4B locus from the TUBB4B-mNeon clonal cell line and confirmed the formation of the correct low-binding lox sequence. The restored sensitivity to blasticidin after excision was then demonstrated in both the Cre-transfected pool and the resulting clonal lines by using selection assays and staining with crystal violet (Fig. 3D). Fluorescent imaging further confirmed the expected localization of the TUBB4B-mNeon endogenous protein across each of the cell lines generated during the editing phases (Fig. 3E). The renewed sensitivity of the APRE-19 TUBB4B-mNeon line was exploited to subsequently tag the histone protein H3C2 with miRFP670nano3-Blast (Fig. 3F,G).

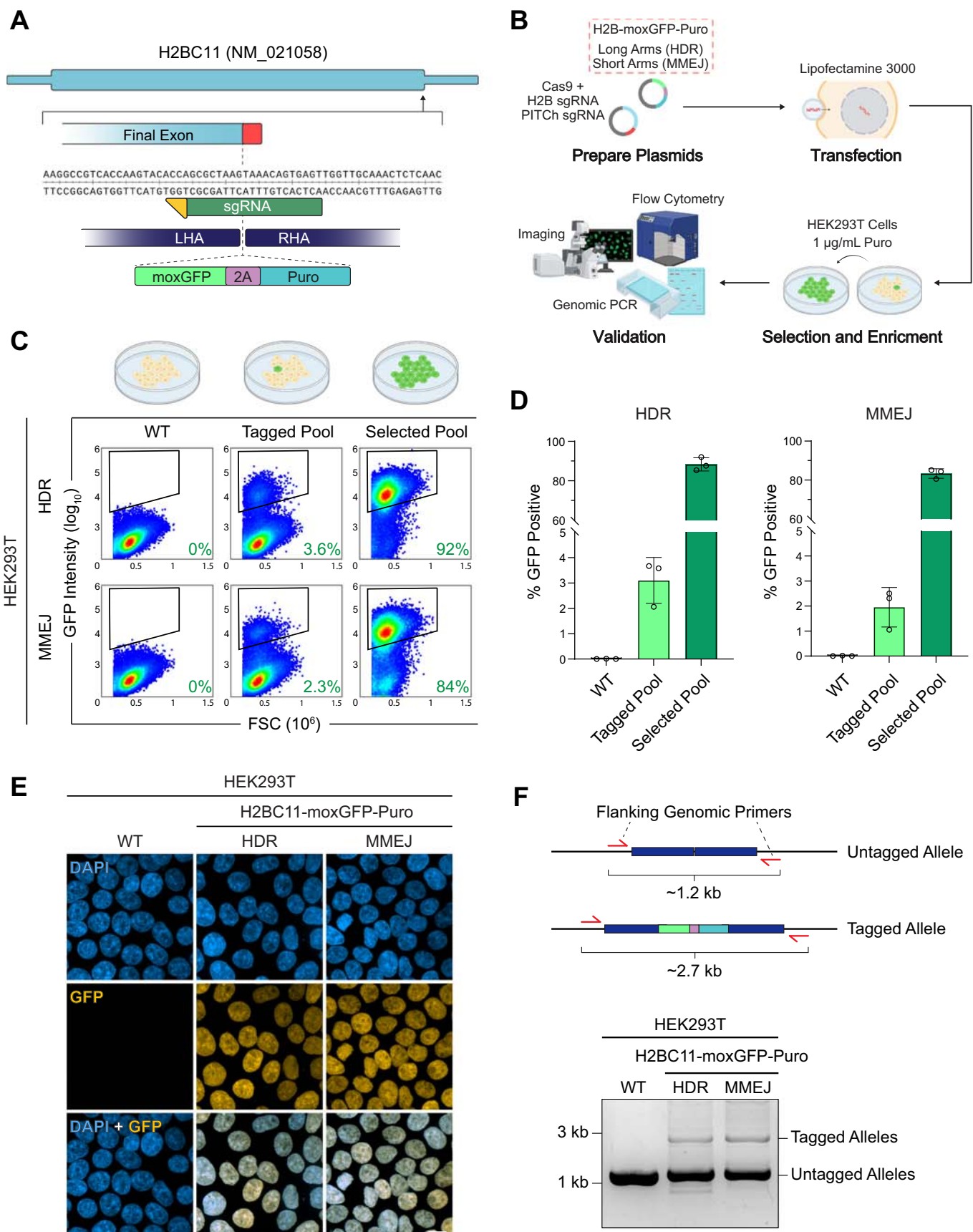

◄ **Figure 2. Editing, enrichment, and validation of fluorescent knock-ins with qTAG cassettes.**

(A) Schematic gene design of the H2BC11 gene displaying the binding of the sgRNA (dark green) and its PAM binding site (yellow), homology arms, and 3′ targeted insertion site. (B) An overview of the strategy used to carry out the endogenous tagging of H2BC11 with a C-terminus encoding insertion of the qTAG-moxGFP-Puro cassette targeting HDR (long homology arms) and MMEJ (short homology arms) repair pathways. (C) Representative flow-cytometry plots depicting 200,000 cells from the following samples: untagged wildtype, the initial tagged H2BC11-GFP-Puro pool, and the final H2BC11-GFP-Puro selected pools of HEK293T cells obtained via targeting the HDR and MMEJ mechanisms. Gates indicates the threshold of fluorescent green cells within the total population as an indicator of positive editing. (D) Flow cytometry quantifications of GFP-positive cells, based on three biological replicates with each measurement encompassing 200,000 cells. A linear axis gap is positioned after 5% and before 60%. Data were mean ± SEM. (E) Representative images of WT, HDR-targeted H2BC11-moxGFP-Puro, and MMEJ-targeted H2BC11-moxGFP-Puro HEK293T cells co-stained with DAPI. Scale bar: 10 µm. (F) An agarose gel displaying the bands produced by genomic PCR with primers located outside the homology arms, probing for locus-specific integration of the qTAG cassette in HEK293T cells. Error bars are mean ± SD from three independent experiments. Source data are available online for this figure.

## Non-fluorescent tagging applications

Examples of gene-tagging with qTAG cassettes containing functional proteomic tags are also presented here. The combination of low editing efficiency and the absence of fluorescent sorting options makes isolating insertions using these tags more challenging. However, coupling them with a selectable marker ensures that the process remains as efficient as our prior demonstrations of fluorescent protein insertion and enrichment. Here, the tagging scheme of the nuclear membrane protein LMNB1 is established and validated using various non-fluorescent tags, including the proximity biotin ligases miniTurbo and ultraID, the targeted degradation tag dTAG, and epitope tags FLAG, HA, and V5 (Fig. 4A). With HAP1 cells, the experimental scheme involving editing, selection, Cre excision, and clonal expansion was followed to generate isogenic tagged lines for proteomic evaluation (Fig. 4B). First, the functionality of two biotin ligases (miniTurbo and ultraID) fused to endogenous LMNB1 was assessed for correct targeting and biotinylation activity. Imaging of HAP1 cells expressing V5-miniTurbo-LMNB1 and V5-ultraID-LMNB1 revealed robust nuclear membrane labeling after treatment with biotin and detection via streptavidin after 1 h (Fig. 4C). Immunoblots using a streptavidin antibody and antibodies against V5 and LMNB1 further confirmed robust biotinylation of proximal proteins within just 10 min, with even greater biotinylation detected after 1 and 12 h (Fig. 4D). Imaging of HAP1 cells expressing V5-dTAG-LMNB1 and co-stained with anti-V5 and anti-LMNB1 antibodies confirmed nuclear membrane localization (Fig. 4E) that was reduced in cells after treatment with dTAG-13, the small molecule responsible for inducing degradation17. Immunoblotting also confirmed the time-dependent degradation of V5-dTAG-LMNB1 in cells treated with dTAG-13 (Fig. 4F). For epitope tagging, the general epitopes FLAG, HA, and V5 were used to tag LMNB1. Immunofluorescence of HAP1 cells stained with epitope-specific antibodies revealed nuclear membrane localization of 3xFLAG-LMNB1, 3xHA-LMNB1, and V5-LMNB1 (Fig. 4G). Additionally, immunoblotting with epitope-specific and target-specific antibodies resulted in robust detection of the endogenous fusion protein, confirming the proper integration of the cassettes and localization of the endogenous fusion (Fig. 4H). As a final point of comparison, we aimed to evaluate the editing efficiencies between fluorescent and non-fluorescent editing cassettes. To achieve this, we generated HDR-targeting donors to insert either a Puro-mStayGold or Puro-dTAG cassette into LMNB1. In this case, we observed reduced localization of LMNB1 in the fluorescent tagging scenario compared to the non-fluorescent tagging scenario (Fig. EV4a,b). However, amplicon sequencing revealed that both tagging methods achieved precise editing when an insertion occurred. The non-inserted alleles had

sustained mutations, with the non-fluorescent scenario exhibiting a larger proportion (Fig. EV4c). For clonal outcomes, a comparable rate of homozygosity was observed in both scenarios (Fig. EV4d,e).

## Compatibility of the qTAG scaffold for the integration of novel tags

We next sought to showcase the adaptability of the qTAG cassette system for future applications by incorporating novel tags. StayGold is a bright green fluorescent protein published in 2022 and is known for its exceptional photostability, which far surpasses that of its contemporary fluorescent proteins (Hirano et al, 2022). However, its inclination to form an obligate dimer to function poses a hindrance in protein fusion applications. Recently, monomeric variants of StayGold were designed to overcome these challenges, including mStayGold (Ando et al, 2024), StayGold-E138D (Ivorra-Molla et al, 2024), and mBaoJin (Zhang et al, 2024). This presents an opportunity for gene tagging applications that facilitate visualization of endogenously tagged proteins, including those expressed at low levels, in long-term live-cell imaging. Due to its enhanced brightness, we opted to pursue a Tag-Only enrichment scheme. In plug-and-play fashion, leveraging the restriction sites within the qTAG cassette, we initiated the process by subcloning the mStayGold sequence into the N- and C-terminus targeting qTAG cassettes (Fig. 5A). Homology arms were subsequently incorporated, and genes tagged with mStayGold were enriched through fluorescence cell sorting. To showcase mStayGold's photostability, the actin gene ACTB was tagged with both mNeon and mStayGold in RPE-1 PAC (−) cells. Homozygous clones of each cell line underwent live-cell imaging with continuous exposure where the normalized fluorescence intensity could be quantified over time (Fig. 5B). Imaging and intensity measurements of these native fusions revealed mStayGold's resilience against photobleaching, exhibiting a discernible difference in intensity within two minutes of acquisition, when compared to mNeon-ACTB, and further maintaining its photostability throughout the entire acquisition period. As an extension of this application, mStayGold-ACTB cells were subjected to live-cell timelapse super-resolution imaging (Fig. 5C). Super-resolution imaging is typically more challenging due to its higher light load, often limiting its application to fixed cell imaging. When combined with endogenous tagging using traditional fluorescent proteins like GFP or mNeon, this challenge is exacerbated, resulting in insufficient signal over time to generate a nice movie. Yet, the enhanced brightness and photostability of mStayGold can reconcile these challenges, allowing for super-resolution structured illumination timelapse microscopy where the leading-edge dynamics in mStayGold-ACTB RPE-1 PAC (−) cells can be observed. Finally, to

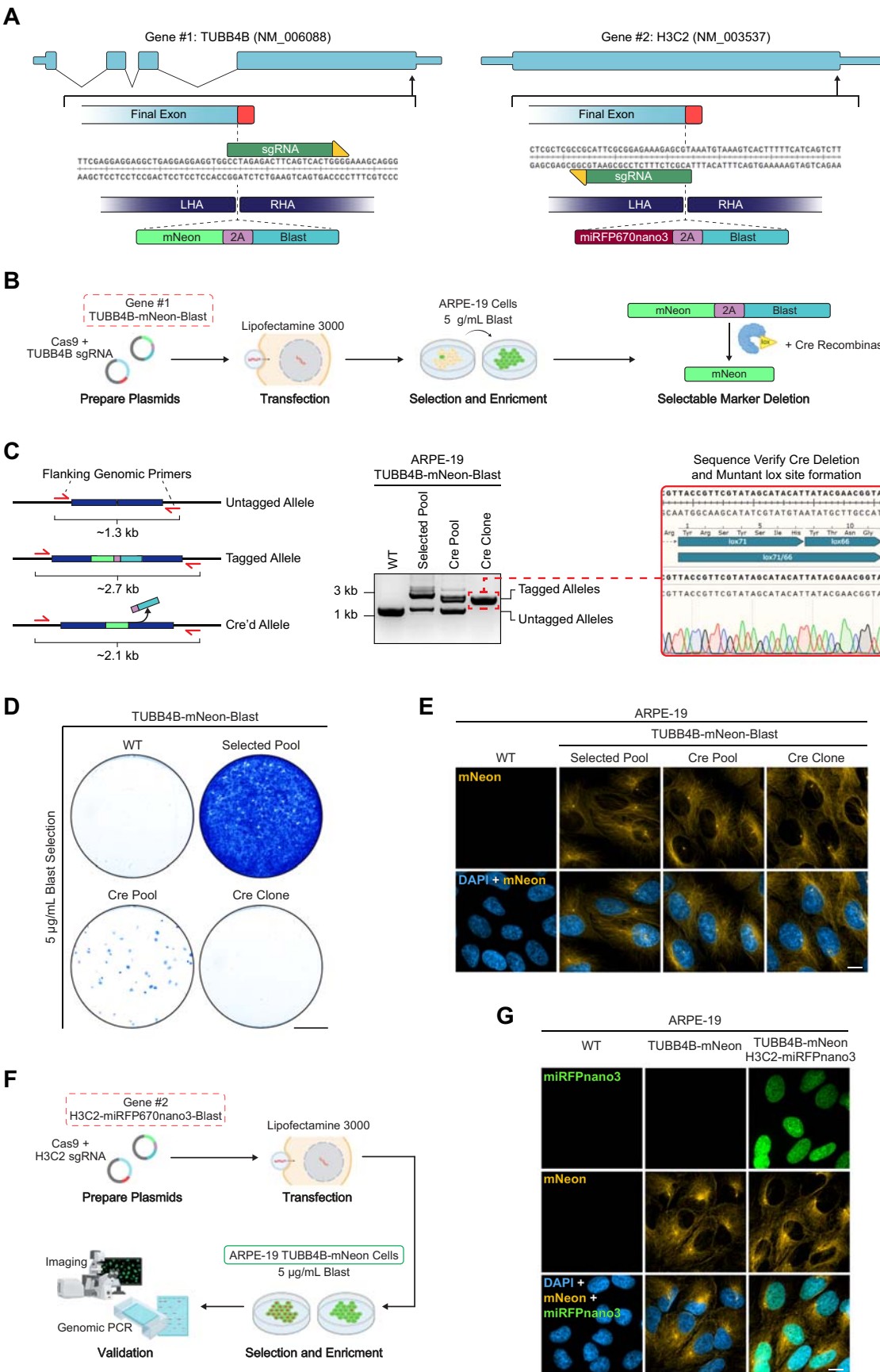

**Figure 3. Selectable marker gene excision via the Cre recombinase and serial endogenous tagging.**

(A) Schematic gene designs of TUBB4B and H3C2. Shown are the location of the sgRNA (dark green) and its PAM binding site (yellow), homology arms, and targeted insertion site. (B) Overview of the strategy used to carry out the tagging of TUBB4B with a C-terminus encoding insertion of a qTAG-mNeon-Blast cassette. (C) An agarose gel displaying bands produced by genomic PCR with primers located outside the homology arms, probing for locus-specific integration of the qTAG cassette and Cre-edited alleles in ARPE-19 cells. Sanger sequencing of a homozygous clonal line displayed accurate excision of the marker gene as displayed by the formation of the inactive lox mutant site post Cre-editing. (D) Representative images of crystal violet-stained plate wells containing WT, selected pool, Cre pool, and Cre clonal cell lines of TUBB4B-mNeon APRE-19 cells selected with blasticidin. Scale bars: 1 cm. (E) Representative images of WT, selected pool, Cre pool, and Cre clonal cell lines of TUBB4B-mNeon APRE-19 cells co-stained with DAPI. Scale bar: 10 μm. (F) Overview of the strategy to carry out subsequent tagging of H3C2 with a C-terminus encoding insertion of qTAG-miRFP670nano3-Blast within the previously edited TUBB4B-mNeon ARPE-19 cells. (G) Representative images of WT, tagged TUBB4B-mNeon, and dual tagged TUBB4B-mNeon and H3C2-miRFP670nano3 ARPE-19 cells co-stained with DAPI. Scale bar: 10 μm. Source data are available online for this figure.

demonstrate the versatility of mStayGold in endogenous molecular fusions, mStayGold qTAG cassettes were used to target genes that illuminate a diverse range of cellular structures (Fig. 5D). These included high-abundance, polymerizing proteins like α-Tubulin to low-abundance proteins localized at the centrosome.

## Exploring different editing applications by modifying the core qTAG cassette structure

Beyond the core qTAG cassette framework, researchers possess the freedom to modify the insertion cargo sequence in creative ways that can allow for alternate enrichment schemes or new tagging scenarios altogether. Here we present three scenarios as examples that expand the utility and applications of the qTAG scaffold.

In situations where the target gene exhibits low or no expression, for example, when gene expression fluctuates during differentiation or reprogramming, accumulating a sufficient amount of selectable markers to confer resistance for the enrichment for integrations can be challenging. The tagging of these genes can be achieved by replacing the "2A" element with the constitutive promoter, such as the short form of the EF1a promoter (EFS) (Fig. 6A).

In our testing, when targeting the cilia gene ARL13B and the centrosome duplication gene PLK4 with a mStayGold-2A-based cassette, selection assays revealed low to no resistant colonies post-selection and crystal violet staining (Fig. 6B,D). However, when targeting these same genes with mStayGold cassettes with EFS-produced selectable markers, many resistant colonies were observed after selection. Imaging of ARL13B-mStayGold-Blast cells in serum-supplemented media and after 72 h of serum starvation confirmed the formation and localization of ARL13B-mStayGold to the cilia when co-stained with an antibody against ARL13B (Fig. 6C—left). The imaging of PLK4-mStayGold-Puro cells also confirms the localization of PLK4-mStayGold to the centrosomes when co-stained with ant-CEP135, a marker for centrioles (Fig. 6E). PLK4 regulates centriole duplication, and its overexpression is known to induce the formation of extra centrosomes (Coelho et al, 2015; Guderian et al, 2010). Thus, by co-staining with anti-CEP135, we could enumerate the centrosomes to determine if extra centrosomes occurred in our edited line. Quantification of centrosome numbers in both our RPE-1 PAC (−) background and our edited PLK4-mStayGold cell line revealed no significant difference between populations with normal centrosome numbers (cells with 1–2 CEP135 foci) and those with amplified centrosome numbers (cells with three or more CEP135 foci). This confirms that the endogenous fusion did not alter PLK4's duplication function, thereby enabling its native study (Fig. 6E–right).

Current methods of creating gene disruptions rely on the infidelity of the NHEJ repair pathway to create random insertions or deletions that disrupt the reading frame. As such, clonal selection and verification is required. Alternatively, a drug selection cassette is inserted at the N-terminus encoding region of a gene to rapidly select for targeted gene knockouts. Here, the qTAG cassette was customized to include a selectable marker terminated with a stop codon, followed by a transcriptional termination sequence in the form of the SV40 polyadenylation signal (see Fig. 6F). To demonstrate the effectiveness of this cassette, we targeted the tumor suppressor protein TP53, performing editing, selection enrichment, and isolating homozygous clones in RPE-1 PAC (−) cells using two different sgRNAs. The cells were then treated with Nutlin-3a, a small molecule inhibitor of MDM2, to induce a TP53-mediated response, resulting in the accumulation of TP53 in TP53-WT cells. Immunoblotting of TP53-WT cells, selected pools of edited TP53-KO-Puro cells, and clonally edited TP53-KO-Puro cell lines using an antibody against TP53 showed TP53 accumulation only in the treated TP53-WT scenario (Fig. 6G). Both the selected pools and clonal cell lines exhibited a loss of TP53 signal under both treatment conditions when using either sgRNA. This finding was further validated through immunofluorescence (Fig. 6H). The loss of the TP53 signal within the selected pool was unexpected, considering that previous editing outcomes had produced clonal homozygosity rates ranging from 0–40%. Upon further analysis of the clonal outcomes from this KO-cassette strategy, it became clear that TP53 sgRNA #1 was highly effective, resulting in over 90% of the clones achieving a homozygous insertion (Fig. EV5a). Although TP53 sgRNA #2 did not yield a high percentage of homozygous edits, many of the clones contained an unknown insertion that effectively abolished TP53 within the pool (Fig. EV5b). As an alternative to this exceptional case, we tested the Golgi gene GOLGA2, which yielded more expected results. Immuno-blotting for GM130 in the WT, GOLGA2-KO-Pool, and GOLGA2-KO-Clone showed a reduced signal in the pool compared to the WT protein, with a complete loss of signal in the clonal cell line (Fig. EV5c). This finding was further supported by imaging, which revealed some loss of Golgi localization in the pool, while no signal was detected in the clone (Fig. EV5d). In terms of clonal outcomes, a modest homozygosity rate was observed compared to TP53, indicating that the effectiveness of the sgRNA plays a critical role in determining editing efficiency and results (Fig. EV5e).

Finally, safe-harbor sites refer to loci in the genome that can accommodate the integration of new genetic elements, ensuring their predictable functionality while minimizing the risk of disrupting normal cellular processes. The AAVS1 safe-harbor site was chosen for the insertion of a modified qTAG cassette to drive gene expression. The design of the homology arms and a portion of the new integration cassette was based on previous strategies used to edit this locus (Hockemeyer et al, 2009; Oceguera-Yanez et al, 2016). This cassette removes the core tag-2A-marker structure and replaces it with a

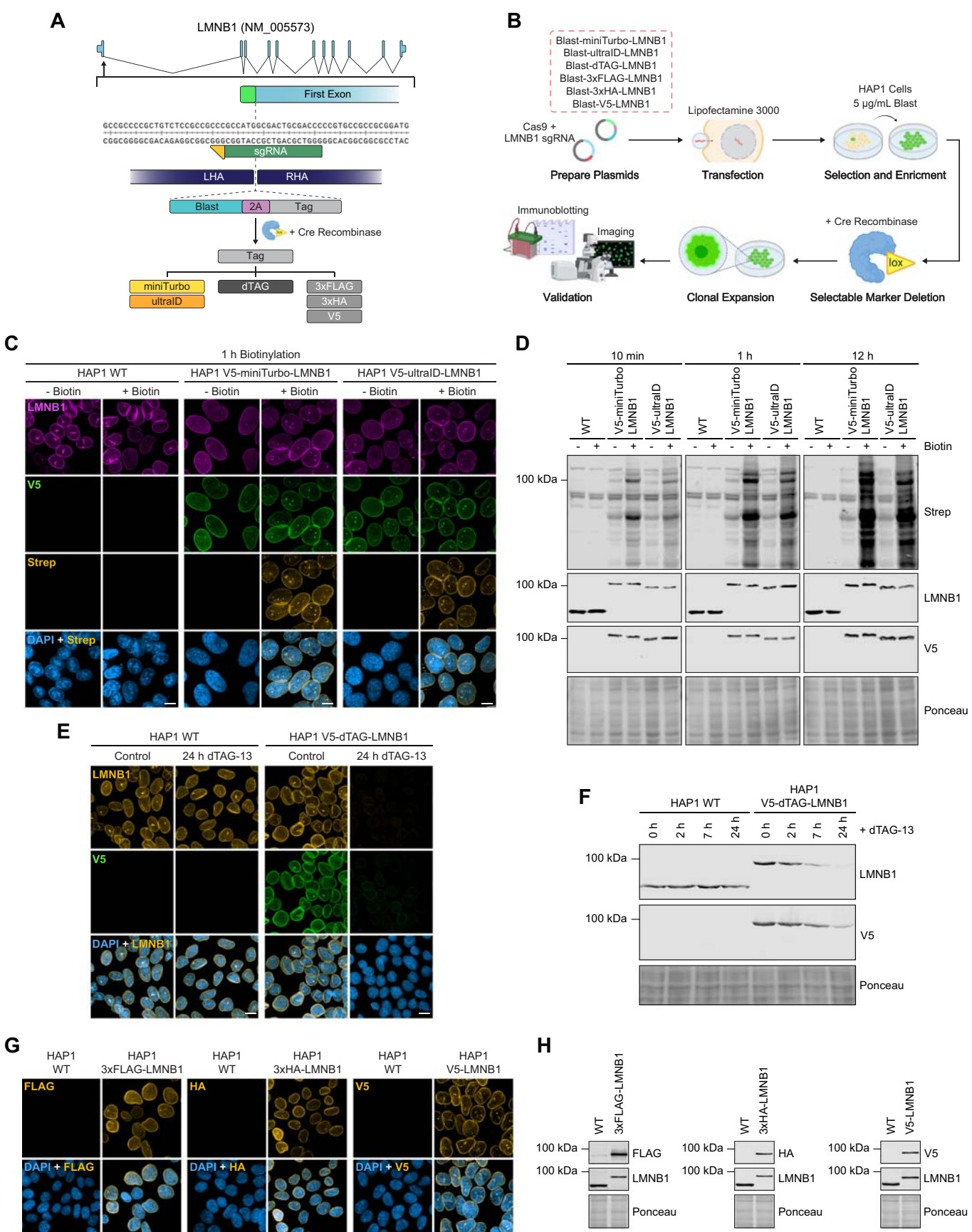

◄ **Figure 4. Non-fluorescent gene qTAG applications and validation.**

(A) Schematic gene design of LMNB1 displaying the binding of the sgRNA (dark green) and its PAM binding site (yellow), homology arms, and targeted insertion site. (B) Overview of the strategy to carry out tagging of LMNB1 with a variety of functional proteomic qTAG cassettes. (C) Representative images of WT, V5-miniTurbo, and V5-ultraID tagged LMNB1 in HAP1 cells. Cells were stained with DAPI and probed with streptavidin in the absence or presence of biotin for 1 h. Scale bars: 10 μm. (D) A representative immunoblot of WT, V5-miniTurbo, and V5-ultraID tagged LMNB1 in HAP1 cells probed with antibodies against streptavidin, V5, and LMNB1 in the absence or presence of biotinylation for 10 min, 1 h, and 12 h. (E) Representative images of WT and V5-dTAG tagged LMNB1 in HAP1 cells stained for DAPI and probed for V5 in the absence or presence of dTAG-13 for 24 h. Scale bars: 10 μm. (F) A representative immunoblot of dTAG-V5 tagged LMNB1 in HAP1 cells probed for V5 and LMNB1 in the absence or presence of dTAG-13 for 2, 7, and 24 h. (G) Representative images of WT, 3xFLAG tagged, 3xHA tagged, and V5 tagged LMNB1 in HAP1 cells stained for DAPI and probed for their respective epitopes. Scale bars: 10 μm. (H) Representative immunoblots of WT, 3xFLAG tagged, 3xHA tagged, and V5 tagged LMNB1 in HAP1 cells with their respective epitope-specific antibodies. Source data are available online for this figure.

selectable marker, a promoter, and a multiple cloning site. This configuration enables the seamless integration of a coding sequence (CDS) of your preference (Fig. 6I). To demonstrate the functionality of this safe-harbor expression cassette, two variants were created using either a PGK or EF1a promoter, with moxGFP cloned into the CDS region. Homology arms targeting AAVS1 were incorporated to facilitate widespread moxGFP expression upon integration. After editing, genomic junction PCR on selected pools confirmed the insertion of the ~4 kb and ~4.7 kb AAVS1 expression cassettes (Fig. 6J). Live-cell imaging of AAVS1-PGK-moxGFP and AAVS1-EF1a-moxGFP selected cells stained with SiR-DNA verified moxGFP expression throughout the cells, with quantification showing over 90% localized fluorescence in both PGK and EF1a pools (Fig. 6K,I).

# Discussion

Ectopic protein expression remains a fundamental approach in experimental research and will likely continue to do so. Ideally, protein behavior would be studied without any modifications, but this is not yet feasible. The most effective current method for investigating protein function involves observing tagged proteins expressed at their native levels. The emergence of the CRISPR/Cas9 system has enabled the widespread use of endogenous tagging over traditional over-expression methods to explore gene function and protein behavior.

Several different strategies have been developed to address the challenges associated with endogenous tagging in cells (Artegiani et al, 2020; Cho et al, 2022; Fueller et al, 2020; Liang et al, 2017; Lin et al, 2019; Perez-Leal et al, 2021; Roberts et al, 2017; Sakuma et al, 2016; Willems et al, 2020; Zhong et al, 2021). These systems incorporate variations like different Cas enzymes, donor repair formats, delivery methods, and post-editing enrichment strategies, but each has its own caveats and limitations. The Allen Institute was among the first to provide a comprehensive set of donor constructs for endogenous tagging in 2017 (Roberts et al, 2017). These donors have been instrumental as one of the initial collections designed to label many major cellular organelles. However, a limitation of this implementation is the adoption of a tag-only approach, which lacks positive selection capabilities and relies solely on fluorescent tagging using older, less performant fluorescent proteins and FACS sorting for selection. The PCR tagging system (Fueller et al, 2020) advocates for simple template production by supplying dsDNA templates in the form of linear PCR cassettes instead of large donor plasmids. These cassettes rely on promoter-based selection and are designed for use with the Cas12 endonuclease. However, this system is limited to C-terminal tagging. Intronic editing systems such as CRISPIE (Zhong et al, 2021) face limitations in their applicability. These systems are most suitable for

genes with short or single amino acid-encoding exons near their 5' or 3' termini. However, such genes are rare exceptions, restricting the overall utility of intronic tagging approaches. When attempting to target insertions between larger exons, these systems risk creating internal fusions that disrupt the native amino acid sequence and functional proteomic domains. An adaptation of this approach involved inserting an intron tagging cassette (Meng et al, 2022) at the N-terminus, offering a clever way to incorporate a promoter-based selection strategy without risking the disruption of the target gene's transcription or the creation of unwanted internal fusion scenarios. More intricate systems such as HITI (Bollen et al, 2022) editing come with the benefits of reducing undesired NHEJ byproducts via concurrent nicking of the genomic locus to integrate the donor target sequence. This approach requires cloning the target sgRNA within the Cas9-nicking construct and embedding sgRNA binding sites in the donor to flank the homology arms, requiring multiple nicking steps to facilitate integration, potentially reducing overall editing efficiencies.

Our primary goal in developing this plasmid system was to create a standardized, widely accessible plasmid scaffold that builds on key principles from previous approaches. This initiative sought to simplify the system for easy use and emphasize modification, allowing researchers to tailor it to their specific experiments. Using the qTAG cassette structure, we demonstrated that these constructs significantly enhance the enrichment of edited outcomes after transfection and selection. In our H2BC11 editing examples, we observed initial enrichment rates of 2.3 to 3.6% when targeting MMEJ and HDR for insertion, respectively. Following selection, the population expressing the fusion increased to 84–92%, indicating a substantial enrichment in the edited pool. We further evaluated localization efficiency, sequence-level outcomes, and clonal integration status to determine the overall editing results. Following transfection and selection enrichment, the tested cell populations frequently exhibited high localization rates, with minimal instances of diffuse or non-localized protein. Amplicon sequencing indicated that alleles in the selected pool that acquired the cassette via HDR had few errors associated with the insertion, whereas those using MMEJ exhibited more errors, and alleles without an insertion were often prone to mutations. While enrichment was highly efficient, clonal analyses revealed that it does not always lead to high rates of homozygosity, likely influenced by sgRNA efficiency. Collectively, these results indicate that aiming for homozygous clonal cell lines is ideal. If only heterozygous cells are available, validations such as sequencing are essential to confirm the integrity of both alleles. Furthermore, we tested the ability of the Cre recombinase to excise the integrated selectable marker gene, as well as showcased a serial tagging scenario in which one selectable marker can be exploited to endogenously tag multiple genes. Lastly, we present scenarios in which we test various non-fluorescent tags, highlighting tagging

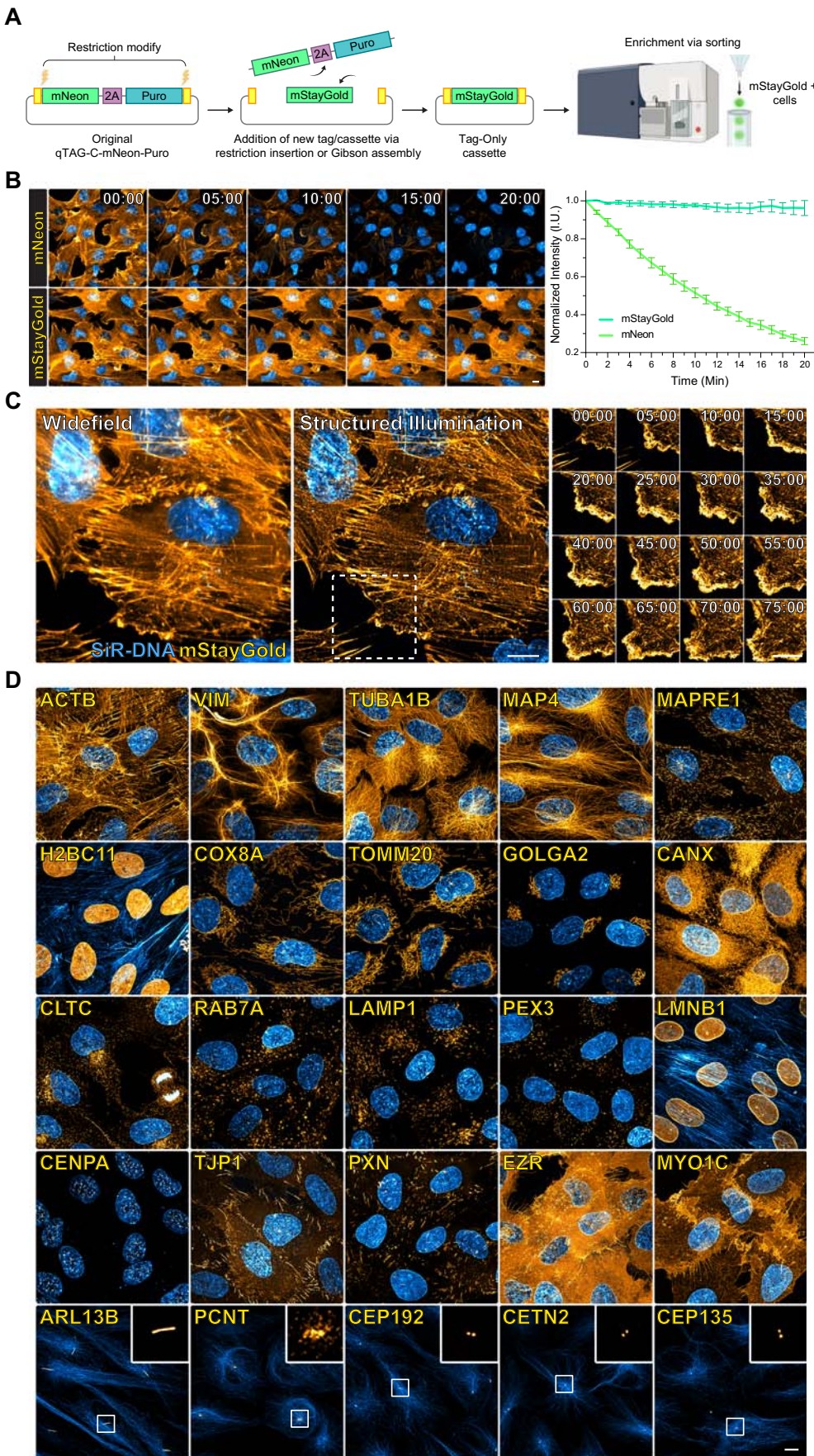

**Figure 5.   Integration of mStayGold in qTAG tagging cassettes.**

(A) A schematic of the modification process to include mStayGold in a Tag-Only knock-in cassette followed by a cell sorting enrichment strategy. (B) Left—Representative live images of N-terminally tagged mNeon-ACTB and mStayGold-ACTB in RPE-1 cells. Cells were live co-stained with SiR-DNA and imaged continuously for 20 min. Scale bar: 10 μm. (B) Right—Normalized intensity measurements of each cell line from three biological replicates, each comprised of four technical replicates. Data were mean ± SEM. (C) Super-resolution timelapse imaging of mStayGold tagged ACTB co-stained with SiR-DNA. (C) Left—A widefield image at the start of the timelapse. (C) Middle—The corresponding structured illumination super-resolution reconstructed image. (C) Right—A timelapse array of images following the highlighted area. Scale bars: 10 μm. (D) Endogenous molecular fusions with mStayGold. Representative images of mStayGold (Orange) tagged to genes highlighting various subcellular compartments, co-stained with SiR (Blue) dyes against Actin, Tubulin, or DNA. Scale bar: 10 μm. Source data are available online for this figure.

situations that are particularly challenging to enrich and isolate without a selection strategy. We validated multiple epitope tags, proximity biotinylation enzymes, and an inducible degron, demonstrating proper localization, verifying endogenous protein levels of the fusions, and assessing the functionality of these fusions. Although we can use methods like genomic PCR, sequencing, immunoblotting, and imaging to confirm the integrity of genetic insertions and protein fusions, the functionality of the tagged fusions remains uncertain and warrants careful examination. Regardless of whether the fusions are overexpressed or endogenously tagged, they are still considered mutant. Therefore, users should always conduct follow-up phenotypic assays to verify that the protein is functioning as intended.

As far as the adaptability of this framework, by incorporating novel tags like mStayGold, we demonstrated its capability for long-term live-cell fluorescence imaging of proteins at native levels. The simplicity of swapping selectable markers and tags with alternative or novel options will empower researchers to develop cassette variants beyond our current offerings. Expansion of proteomic tagging could encompass alternative degron tags, HaloTag, SnapTag, and ALFA tag, while fluorescent tags with emission spectra not covered in our collection (cyan, yellow, near-infrared, photo-activation, long stokes-shift) could be easily accommodated. Furthermore, this framework opens avenues for exploring the insertion of any cargo sequence within the "Tag" region of these cassettes, potentially facilitating the attachment of sequence elements such as signal peptides, subcellular targeting domains, cleavage signals, and binding domains as endogenous modifications to manipulate gene function or localization.

However, within the standardized structure of the provided qTAG cassettes, two limitations do exist. One drawback is the reliance on transcriptional linkage of the target gene to the selectable marker with the 2A sequence. This requires that the target GOI be expressed at high enough levels in the cell to produce enough antibiotic resistance for efficient selection. Also, while 2A peptide sequences have garnered mass adoption in molecular biology to achieve co-translation, their cleavage or 'skipping' efficiency is not always consistent in all cellular backgrounds. This can result in a minor amount of read-through (Kim et al, 2011; Liu et al, 2017), resulting in fusions between the tagged GOI and the selectable marker. The effects on transcript levels, particularly concerning the positioning of the 2A peptide, have also been reported, highlighting the importance of Cre excision in the qTAG system (Liu et al, 2017). To address these limitations and enhance flexibility, we introduced separate cassettes where the tag region terminates with a stop codon, and the 2 A sequence is replaced with an EFS promoter. This modification allowed the selection of edits independent of the target gene's transcript levels and circumvents the constraints associated with 2A peptide sequences. Thanks to the constitutive expression of the selectable marker, even genes that are expressed at low levels or transcriptionally silent could be potentially edited. This could be particularly useful in cases where the

tracking of fate-based gene expression is important, such as during the differentiation or reprogramming of cells, where the activation of fate-specific transcription factors often occurs. By silently tagging these genes at their C-terminus encoding region, either directly with a fluorescent protein sequence or with a 2A-fluorescent protein sequence, a cell line with an intrinsic disposition marker can be generated. Upon reaching the particular fate where the targeted gene would be expressed, the tagged transcription factor would fluoresce if directly tagged or illuminate the entire differentiating/reprogramming cell if tagged with a 2A-Fluorescent protein sequence, effectively serving as a transcriptional biosensor for cellular fate.

Beyond the core tag-2A-marker structure, we further modified the payload insertion region to explore two alternative tagging scenarios: facilitating selectable gene knockouts and enabling safe-haven exogenous expression. In the context of selectable knockout cassettes, there are situations where transfectability, poor editing efficiency, and/or clonal expansion is impractical due to limited replicative potential (such as in primary cells). In such instances, employing a selectable knockout scenario significantly shortens the timeline compared to traditional methods involving clonal indel validation. Although considerations regarding ploidy and copy number integration persist in both scenarios, the availability of a selectable enrichment option alleviates the challenges associated with factors like transfection efficiency and sgRNA binding efficiency. This was illustrated by the significant reduction in the target protein observed even within our selected pool of TP53-KO cells.

Our last alternative tagging approach focused on tagging the safe-harbor site AAVS1 with an expression cassette. This represents the most extensive alteration of the cassette structure, introducing multiple new genetic elements to fulfill a completely different purpose in accommodating exogenous expression. Common methods for achieving permanent integration of expression cassettes involve either random integration of a plasmid post-transfection or the use of lentivirus systems. However, these methods either depend on chance or entail specific considerations regarding viral culture, and result in random integration into the genome. Conversely, the preference often lies in integrating expression cassettes into predetermined genomic loci, a feat typically accomplished via the FLP-In system (Szczesny et al, 2018). However, this process necessitates specialized FRT reagents, cell lines, and is time-intensive. The safe-harbor overexpression tagging approach addresses many of these challenges by facilitating quick, permanent, and site-specific integration for the expression of a target CDS.

In summary, we have detailed the development and application of a versatile endogenous tagging platform. Engineered with the aim of making CRISPR-mediated tagging more accessible, the qTAG library of plasmids incorporate various essential characteristics that enhance usability and adaptability, serving as an open framework for researchers to tailor towards their own experiments.

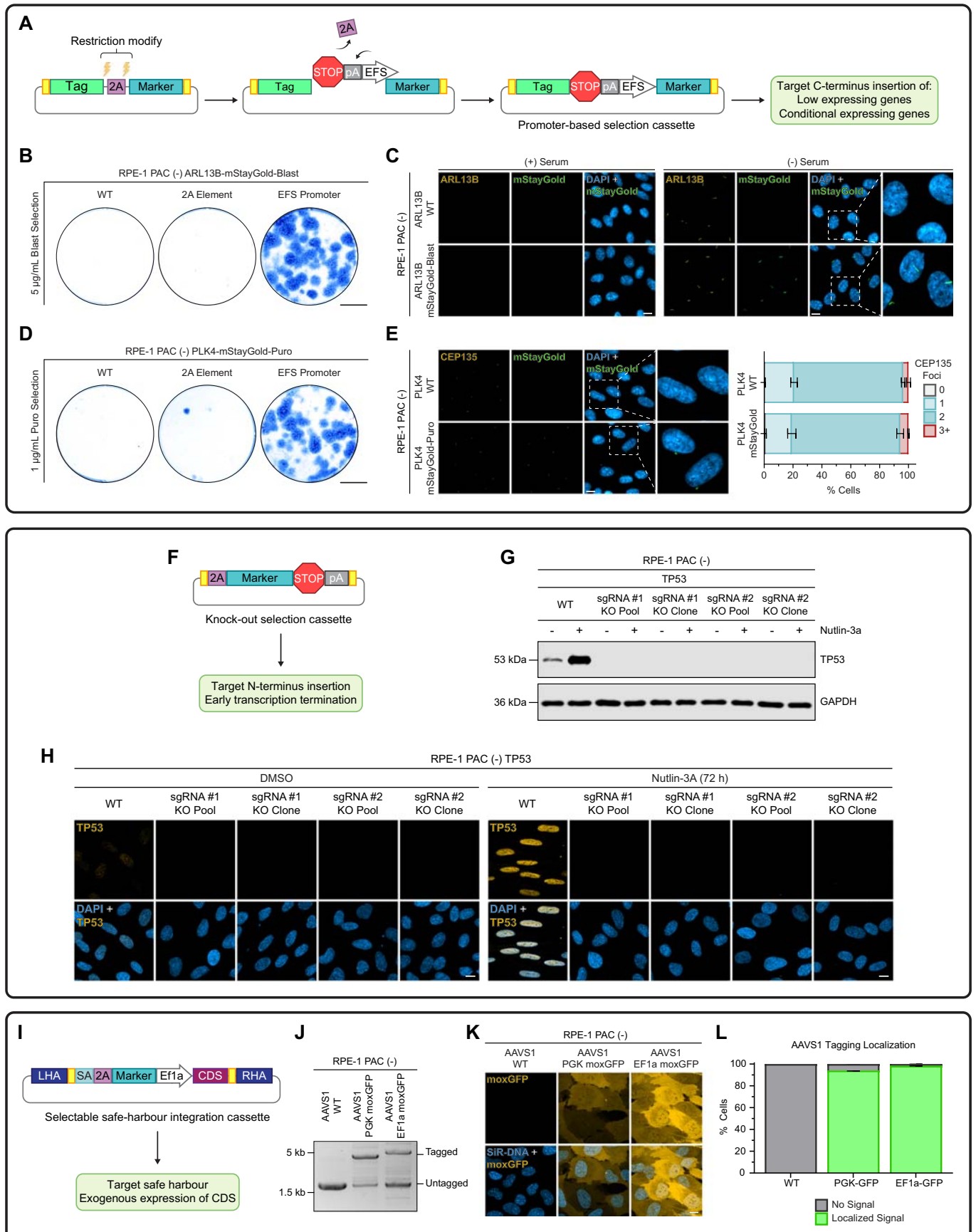

**Figure 6. Generating alternative cassette designs for endogenous tagging and beyond.**

(A) A schematic of the modification process to convert the cassette from a multicistronic 2A-based approach to a promoter-based approach for the enrichment of integrations. pA polyadenylation signal, EFS EF1a short promoter. (B) Representative images of crystal violet-stained plate wells containing WT-ARL13B, initially tagged 2A-based ARL13B-mStayGold, and initially tagged promoter-based ARL13B-mStayGold RPE-1 PAC (−) cells selected with blasticidin for 7 days. Scale bar: 1 cm. (C) Representative images of WT and endogenously tagged, promoter-based ARL13B-mStayGold. Cells were stained with DAPI and probed with anti-ARL13B in the absence or presence of serum starvation for 72 h. Scale bars: 10 μm. (D) Representative images of crystal violet-stained plate wells containing WT-PLK4, initially tagged 2A-based PLK4-mStayGold, and initially tagged promoter-based PLK4-mStayGold RPE-1 PAC (−) cells selected with puromycin for 7 days. Scale bar: 1 cm. (E) Left—Representative images of WT and endogenously tagged, promoter-based PLK4-mStayGold. Cells were stained with DAPI and probed with anti-CEP135. Scale bar: 10 μm. (E) Right—a quantification of centrosome number within the WT and promoter-based PLK4-mStayGold pools. >100 cells were quantified in each of the three biological replicates for each condition. Data were mean ± SEM. (F) A schematic of a cassette that expresses a selectable marker followed by the induction of transcriptional termination allowing for selectable knockouts if integrated at the N-terminal encoding region of a gene. (G) A representative immunoblot of RPE-1 PAC (−) WT-TP53 cells, TP53-KO-Puro selected pools of cells using two sgRNAs, and TP53-KO-Puro selected clonal cell lines probed with antibodies against TP53 and GAPDH in the absence or presence of 10 μM Nutlin-3a for 72 h. (H) Representative images of RPE-1 PAC (−) WT-TP53 cells, a TP53-KO-Puro selected pools, and TP53-KO-Puro selected clones probed with anti-TP53 in the absence or presence of 10 μM Nutlin-3a for 72 h. Scale bars: 10 μm. (I) A schematic of a cassette that allows for the integration at the safe-harbor AAVS1 loci, allowing for exogenous expression. SA splice acceptor, CDS coding sequence. (J) Genomic PCR outside the homology arms probing for locus-specific integration of the AAVS1 safe-harbor moxGFP in RPE-1 PAC (−) cells. (K) Representative images of RPE-1 PAC (−) WT-AAVS1 cells, AAVS1-PGK-moxGFP-puro selected cells, and AAVS1-EF1a-moxGFP-puro cells co-stained with DAPI. Scale bar: 10 μm. (L) Quantification of cells expressing moxGFP within the WT and selected AAVS1 edited cell lines expressing moxGFP. >500 cells were quantified in each of the three biological replicates for each condition. Data were mean ± SEM. Source data are available online for this figure.

# Methods

### Reagents and tools table

| Reagent/resource | Reference or source | Identifier or catalog number |
|---|---|---|
| **Experimental models** | | |
| HAP1 cells (RRID: CVCL_Y019) | Horizon Genomics | C631 |
| HEK293T cells (RRID: CVCL_0063) | ATCC | CRL-3216 |
| ARPE-19 Cells (RRID: CVCL_0145) | ATCC | CRL-2302 |
| RPE-1 cells (RRID: CVCL_4388) | ATCC | CRL-4000 |
| U-2 OS cells (RRID: CVCL_0042) | ATCC | HTB-96 |
| H9 cells (RRID: CVCL_9773) | WiCell | WA09 |
| **Recombinant DNA** | | |
| pX330-U6-Chimeric_BB-CBh-hSpCas9 | Addgene | https://www.addgene.org/42230/ |
| pX330-BbsI-PITCh | Addgene | https://www.addgene.org/127875/ |
| pET-NLS-Cas9-6xHis | Addgene | https://www.addgene.org/62934/ |
| **Antibodies** | | |
| Streptavidin | Li-Cor Biosciences | 926-68079 |
| LMNB1 | Abcam | AB16048 |
| FLAG | Cell Signaling | 14793 |
| HA | BioLegend | 901501 |
| V5 | Life Technologies | R960-25 |
| TP53 | Santa Cruz | sc-126 |
| ARL13B | Santa Cruz | sc-515784 |

| Reagent/resource | Reference or source | Identifier or catalog number |
|---|---|---|
| CEP135 | produced in-house | |
| GM130 | BD Biosciences | 610822 |
| GAPDH | MilliporeSigma | G9545 |
| **Oligonucleotides and other sequence-based reagents** | | |
| GeneArt Strings | Thermo Fisher Scientific | |
| **Chemicals, enzymes, and other reagents** | | |
| DMEM | Gibco | 11965084 |
| DMEM/F-12 | Gibco | 11320033 |
| CloneR | StemCell Tech | 05889 |
| mTeSR1 | StemCell Tech | 100-0276 |
| McCoy's 5 A medium | Gibco | 16600082 |
| 0.5% crystal violet | MilliporeSigma | 46364 |
| TrypLE | Thermo Fisher Scientific | 12605028 |
| Lipofectamine 3000 | Invitrogen | L3000008 |
| Cloning Cylinders | MiliporeSigma | CLS31666 |
| Silicone Grease | MiliporeSigma | 85410 |
| Cell Strainer | Corning | 352235 |
| QuickExtract | Lucigen | LGN-QE09050 |
| QIAquick Gel Extraction Kit | Qiagen | 28706 |
| DTT | Amresco | 0281-25 G |
| PonceauS | MilliporeSigma | P7170 |
| protease inhibitor cocktail | Sigma-Aldrich | P8340 |
| Prolong Gold | Thermo Fisher Scientific | P36930 |
| Nutlin-3a | Cayman Chemical | 1858 |
| Plasmid Plus Midi Kit | QIAGEN | 12945 |
| QIAquick PCR Purification | QIAGEN | 28106 |
| QIAprep Spin Miniprep | QIAGEN | 27106 |

| Reagent/resource | Reference or source | Identifier or catalog number |
|---|---|---|
| DMSO | Sigma-Aldrich | D8418 |
| FastDigest BpiI or BbsI | Thermo Fisher Scientific | FD1014 |
| FastAP | Thermo Fisher Scientific | EF0651 |
| T4 Polynucleotide Kinase | New England Biolabs | M0201S |
| T4 DNA Ligase Kit | New England Biolabs | M0202S |
| Quick Ligation Kit | New England Biolabs | M2200S |
| Gibson Assembly Master Mix | New England Biolabs | E2611L |
| DH5α competent cells | Invitrogen | 18258012 |
| Ampicillin | Sigma | A5354 |
| Matrigel | VWR | CA89050-192 |
| Fetal Bovine Serum | Gibco | A4736301 |
| phosphate-buffered saline | Gibco | 14190250 |
| Trypsin-EDTA-0.25% | Gibco | 25200056 |
| GlutaMAX | Gibco | 35050061 |
| Penicillin/streptomycin | Gibco | 15140122 |
| Neon Transfection Kit | Thermo Fisher Scientific | MPK10096, MPK5000 |
| Opti-MEM Reduced Serum Medium | Gibco | 31985070 |
| Puromycin | InvivoGen | ant-pr-1 |
| Blasticidin | InvivoGen | ant-bl-1 |
| Zeocin | InvivoGen | ant-zn-1 |
| AgeI-HF | New England Biolabs | R3552L |
| BamHI-HF | New England Biolabs | R3136S |
| EcoRI-HF | New England Biolabs | R3101S |
| HindIII-HF | New England Biolabs | R3104S |
| KpnI-HF | New England Biolabs | R3142S |
| MluI-HF | New England Biolabs | R3198L |
| NheI-HF | New England Biolabs | R3131L |
| XbaI | New England Biolabs | R0145L |
| XhoI | New England Biolabs | R0146S |
| XmaI | New England Biolabs | R0180L |
| RedSafe | FroggaBio | 21141 |
| HiScribe T7 High Yield RNA Synthesis Kit | New England Biolabs | E2040L |
| RNAClean XP | Beckman Coulter | A63987 |
| Bradford Protein Assay Kit | Thermo Fisher Scientific | 23200 |
| **Software** | | |
| NIS-Elements 5.42 | Nikon | |
| ImageJ (Fiji) | | |
| Snapgene 7.1.2 | | |
| FlowJo 10.10 | BD Biosciences | |
| CHOPCHOP v3 | https://chopchop.cbu.uib.no/ | |

| Reagent/resource | Reference or source | Identifier or catalog number |
|---|---|---|
| Cellprofiler | | |
| BioRender | | |
| Prism 10.2 | GraphPad | |
| **Other** | | |

## qTAG system construct development

The initial structures of the C- and N-terminus targeting cassettes were sourced from Nabet et al (Nabet et al, 2018). These sequences were arranged from 5′ to 3′ to include a protein tag sequence in frame with a P2A peptide sequence, succeeded by a selectable marker sequence (with the tag and selectable marker sequence switched for the N-terminal targeting cassettes). Unique restriction sites were introduced next to every sequence element to allow for easy variant cloning. Prior to synthesizing codon-optimized gene fragments for the initial C- and N-terminus cassettes (GeneArt Strings, Thermo Fisher), uniform cloning sequences were introduced around the cassette and lox sites to flank the selectable marker region. Following this, the cassettes were cloned into a pUC19 backbone. Variants of the cassette were produced via restriction insertion cloning, using specific enzymes to the qTAG cassette and the insertion sequences (AgeI-HF, BamHI-HF, EcoRI-HF, HindIII-HF, KpnI-HF, MluI-HF, NheI-HF, XbaI, XmaI, NEB), followed by ligation (T4 DNA Ligase Kit, M0202, NEB) of the resulting products at a 1:3 molar ratio of vector to insert. Table EV1 presents the sources of the tag sequences used. Separately, Cre expression plasmids were generated to complement the qTAG cassettes. Fragments containing an Ef1a promoter, Cre recombinase CDS, P2A element, puromycin selectable marker CDS, and a polyA signal were synthesized, followed by Gibson assembly (Gibson Assembly Master Mix, E2611L, NEB) at a 1:3 molar ratio of vector to fragment for directional cloning into a pUC19 backbone. All presently generated variants of qTAG cassettes and Cre plasmids are listed in Table EV4.

## Gene knock-in design, CRIPSR, and repair construct assembly

CHOPCHOP v3 (Labun et al, 2019) (https://chopchop.cbu.uib.no/) was our primary sgRNA design tool because of its user-friendly graphical interface, which provides clear visualization of genomic loci and sgRNA binding sites. The selection of suitable sgRNAs for gene tagging design relied on several criteria. First, sgRNAs were locally filtered, being restricted to sgRNAs binding positions within ~30 bp of the 5′ or 3′ ends of a gene, depending on whether tagging was planned for the N- or C-terminus, respectively. Beyond location, predicted on-target and off-target scores were considered when choosing and ranking potential sgRNAs. To produce the sgRNAs within cells, we employed either plasmids encoding Cas9 and target sgRNA or delivered the Cas9 protein and sgRNAs directly as a ribonucleic protein complex (RNP). For plasmid-based delivery, the original Zhang protocol for SpCas9 plasmids (https://www.addgene.org/crispr/zhang/) was used to clone and deliver our Cas9 and sgRNA (Cong et al, 2013). We particularly used either

pX330 (https://www.addgene.org/42230/) or the MMEJ PITCh-primed variation of px330, the px330-PITCh plasmid (https://www.addgene.org/127875/) (Nakamae et al, 2017; Sakuma et al, 2016). For RNP-based delivery, Cas9 was expressed from a pET-based T7 promoter-containing plasmid (https://www.addgene.org/62934/) in BL21 bacteria (Zuris et al, 2015). The purification of Cas9 was carried out from a previously described method (Anders and Jinek, 2014). To produce the sgRNA, a one-step PCR assembly method (Hu et al, 2019) was performed to generate the guide DNA basis. In-vitro transcription (HiScribe T7 High Yield RNA Synthesis Kit E2040L, NEB) from the DNA was then carried out, followed by RNA purification (RNAClean XP, Beckman Coulter).

For HDR-targeted qTAG repair constructs, 500–1000 bp of homologous genomic sequence flanking the predicted sgRNA cut site was either directly amplified from the genome via PCR or synthesized as gene fragments. The homology arms were generated with specific flanking cloning sequences outlined in Table EV2 that are homologous to those on the qTAG cassettes. To ensure that the sgRNA sequence did not overlap with the donor construct, sgRNAs were chosen such that they would inherently disrupt the sgRNA sequence upon insertion. In situations where this was not feasible, PAM blocking or sgRNA blocking mutations were introduced during homology arm synthesis to prevent Cas9 cleavage of the donor construct upon entry into the nucleus or re-cleavage of the genomic insertion post-integration. For the final two fragments, the backbone fragment involved digesting the pUC19 backbone using EcoRI and HindIII enzymes, whereas the designated qTAG cassette intended for insertion was amplified using the cassette-specific primers listed in Table EV2. These four fragments of digested pUC19 backbone, left homology arm, amplified qTAG cassette, and right homology arm were then directionally cloned using Gibson assembly at a 1:3:3:3 molar ratio to form the final qTAG HDR-targeting repair construct.

For MMEJ-targeting qTAG repair constructs, flanking homology of 10–20 bp around the sgRNA cut site was directly added to the cassette amplification primers used to amplify the preferred qTAG cassette. PITCh sgRNA sequences were also included in the primers following the homology sequences to allow for cassette release upon cleavage. The qTAG MMEJ primer design scaffold is outlined in Table EV2. Following the PCR process with the MMEJ primers and the selected qTAG cassette, the resultant cassette fragment underwent digestion and was subsequently restriction-cloned into a pUC19 backbone.

In both the HDR and MMEJ-targeting cases, the repair constructs were purified (Plasmid Plus Midi Kit, 12945, QIAGEN) and sequence verified via Sanger sequencing using the sequencing primers outlined in Table EV2 prior to introduction into cells. Source gene designs of all of the genes targeted in this paper are highlighted in Table EV3 with their corresponding qTAG repair variants summarized in Table EV4. Additionally, supplementary example annotated sequence data files are provided for a more detailed view of knock-in design and the resultant qTAG repair constructs (Dataset EV1).

## Cell culture

The human cell lines used in this study included: HAP1 cells (Horizon Genomics, cat. No. C631, RRID: CVCL_Y019), HEK293T cells (ATCC, cat. No. CRL-3216, RRID: CVCL_0063), ARPE-19 cells (ATCC, cat. No. CRL-3216, RRID: CVCL_0145), RPE-1 cells (ATCC, cat. No. CRL-4000, RRID: CVCL_4388), U-2 OS cells (ATCC, cat. No. HTB-96, RRID: CVCL_0042), H9 cells (WiCell, cat. WA09, RRID: CVCL_9773). RPE-1 with a knocked-out PAC (puromycin N-acetyltransferase) gene was obtained as a gift from Dr. Daniel Durocher's laboratory. DMEM (11965084, Gibco) supplemented with 10% FBS was used to culture HAP1 and HEK293T and cells. For U-2 OS cells, McCoy's 5A medium (16600082, Gibco) supplemented with 10% FBS was used. DMEM/F-12 (11320033, Gibco) supplemented with 10% FBS was used to culture ARPE-19 and RPE-1 PAC (−) cells. mTeSR1 (100-0276, StemCell Tech) and CloneR (05889, StemCell Tech) were used for regular and clonal culture H9 cells, respectively. All cells were grown in a tissue culture incubator with humidity at 37 °C and 5% $CO_2$.

## Transfection and editing

Chemical delivery of editing components via Lipofectamine 3000 into tissue culture cell lines began with seeding approximately 500,000 cells (or enough to reach 70–80% confluency the next day) per well in a six-well plate. Each experiment included a well designated for transfection and selection controls, positioned alongside the well containing the gene-tagging target. The subsequent day, co-transfection of pX-CRISPR and qTAG repair plasmids was performed at a 1:1 ratio using 4 μg of total DNA (2 μg cut + 2 μg repair) in Opti-MEM Reduced Serum Medium (31985070, Gibco) with Lipofectamine 3000 (L3000008, Invitrogen). Media replacement occurred the next day and an editing period of 72 h post-transfection was allowed prior to selection enrichment. It was crucial to transfer cells to a larger vessel (six-well >10 cm dish) if confluency was reached to ensure proper cell growth and prevent contact inhibition.

The delivery of editing components via Neon electroporation (MPK10096, MPK5000, Thermo Fisher) and RNPs into H9 human stem cells involved several steps. Initially, H9 cells were dissociated into single cells using TrypLE (12605028, Thermo Fisher) and then collected via centrifugation and resuspended in mTeSR1. RNP complexes were assembled on ice by incubating purified Cas9 protein with the sgRNA, followed by addition to cells at a cell concentration of $1 \times 10^7$ cells/mL in 10 μL. Electroporation was carried out using specific parameters (1100 V, 30 ms, 1 Pulse) using 10 μL Neon tips, and cells were rapidly transferred to media containing CloneR in a Matrigel-coated well. The following day, media replacement was performed, and an editing period of up to 72 h was allowed before selection enrichment.

## Enrichment of edits via mammalian antibiotic selection or FACS

After editing, the culture media was replaced with media supplemented with the mammalian antibiotic corresponding to the tagging cassette used. A range of working concentrations was tested for ARPE-19, RPE-1, HEK293T, HAP1, U-2 OS, and H9 cells. To enrich for knocked-in cells in human tissue culture cell lines, concentrations of 5–10 μg/mL Blasticidin, 0.5–1 μg/mL Puromycin, and 100–400 μg/mL Zeocin were used, with complete selection being observed in 4–7 days, 2–3 days, and 10–15 days, respectively. For H9 stem cells, concentrations of 3 μg/mL Blasticidin, 0.5 μg/mL Puromycin, and 25–50 μg/mL Zeocin were

used, with complete selection being observed in 7–8 days, 2–3 days, and 10–15 days, respectively. Once all untransfected control cells had been eradicated, the resultant resistant cells represented the initial enriched tagged pool. It was critical to note that the initial number of remaining cells in some cases were limited, depending on factors such as guide or transfection efficiency. In such cases (Certain gene designs including PLK4, ARL13B, and TOMM20), an additional 2–5 days were allotted for the growth of potential resistant colonies. If it wasn't feasible to incorporate mammalian selectable markers or if a Tag-Only approach was employed, as seen in the mStayGold experiments, edited cell enrichment was alternatively performed using FACS (BD FACS Aria Fusion). Fluorescence thresholds and gates were based on the negative-control, unedited population.

## Cre selectable marker deletion

Cre selectable marker excision was carried out after the generation of a selected pool of edited cells. 500,000 of the target tagged cell line was seeded per well in a 6-well plate to achieve ~80% confluency the following day. Cre recombinase delivery involved transfection of the chosen Cre encoding plasmid, Cre-Puro, in the case of Fig. 3, using Lipofectamine 3000 as previously described. The following day, the selection of cells commenced either through fluorescence or puromycin resistance, dependent on the Cre plasmid used, to enrich for cells expressing the Cre recombinase, resulting in the generation of a tagged pool of cells with the integrated selectable marker being removed. The precaution was taken to remove the selection immediately after the elimination of the control cells to avoid enriching for permanent integrations of the Cre plasmid itself.

## Clonal cell line generation

To isolate clones, ~350–500 cells from the selected pool of edited cells were seeded into a 15 cm dish. About 7–10 days were allotted for the formation of distinct colonies. Cellular clones were then isolated using cloning cylinders (CLS31666, MilliporeSigma) coated in sterile silicone grease (85410, MilliporeSigma), where the clones could be trypsinized and transferred for further propagation.

## Flow cytometry

Flow analysis of the edited fluorescent populations was carried out to assess tagging efficiency and enrichment using the qTAG cassettes. Control, initial tagged pools, and selected pools of cells were resuspended in a PBS, 2% FBS, 0.5 mM EDTA buffer, and filtered through a cell strainer (352235, Corning) to make a single cell suspension. Samples were then subjected to analysis on an Attune NxT flow cytometer (Thermo Fisher), where size and fluorescence thresholds and gates were based on the negative-control, unedited population. For each sample, ~200,000 events were recorded across three biological replicates.

## Selection assay

For selection assay experiments, 200,000 cells previously transfected with editing reagents were seeded in six-well plates. The next day medium was removed and medium containing the indicated selection was added. The media was refreshed every 3–4 days to

ensure continued selection. After 7 days, plates were rinsed once with PBS and fixed and stained with 0.5% crystal violet (46364, MilliporeSigma) in 20% methanol for at least 20 min. Plates were washed extensively with water, dried, and scanned.

## Allelic screening with genomic junction PCR

To check for the allelic integration status of the cassettes, genomic DNA was extracted from 10,000–50,000 target cells by resuspending them in a QuickExtract DNA Extraction Solution (LGN-QE09050, Lucigen). PCR amplification of the targeted junction from the wildtype and tagged cells was performed using primers designed to anneal outside of the homology arms of the specific gene. The resulting PCR products were analyzed on an agarose gel, where it was expected that the size of the amplicon from the tagged alleles would be proportional to the qTAG insertion cassette size plus the wildtype amplicon size. For further validation of framing and the absence of mutations, the amplicons were gel purified using the QIAquick Gel Extraction Kit (28706, QIAGEN) and subjected to Sanger sequencing.

## Amplicon sequencing

DNA from both untransfected controls and transfected, selected pools was extracted and subjected to PCR using primers located outside the left and right homology arms (Dataset EV2). The total PCR product was purified and sequenced by Plasmidsaurus using Oxford Nanopore Technology with custom analysis and annotation. Raw reads from the fastq file were analyzed with 2FAST2Q (Bravo et al, 2022) (https://github.com/afombravo/2FAST2Q), using the command "python -m fast2q -c --mo EC --ph 0 --us xxxx --qsu 1 --l yy," where "xxxx" is the search sequence and "yy" is the search length (see Dataset EV2). For each targeted gene, the 5′ and 3′ insertion sites were analyzed using one forward and one reverse search sequence. Sequences with more than 20 counts were considered for further analysis. At each junction, sequences matching the expected result were classified as "wildtype," while all others were deemed "mutant." For each PCR reaction, wildtype and mutant reads corresponding to untagged and tagged products were analyzed separately to calculate the percentage of wildtype versus mutant reads.

## Immunoblot analyses

For immunoblot analyses, cells were seeded into six-well plates and cultured until confluency was reached. The indicated cells were then scraped and lysed on ice for 10 min with RIPA buffer (150 mM NaCl, 1% Triton X-100, 0.5% sodium deoxycholate, 0.1% SDS, 50 mM Tris-HCl at pH 8.0, 10 mM NaF, 1 mM Na3VO4, 1 mM EDTA, and 1 mM EGTA) containing a protease inhibitor cocktail (Sigma-Aldrich, P8340) and centrifuged at 12,000 r.p.m. for 10 min at 4 °C. To quantitate the protein, a fraction of the supernatant was subjected to a Bradford assay (Bradford Protein Assay Kit, 23200, Thermo Fisher). The rest of the supernatant was mixed with 4x SDS sample buffer (250 mM Tris-HCl at pH 6.8, 8% SDS, 40% glycerol, and 0.04% bromophenol blue) and 10 mM DTT (Amresco, 0281-25 G). The mixtures were boiled for 5 min. To ascertain the presence of the endogenous protein or its endogenous tag, 10 ug of protein of each sample was loaded in 10%

SDS–polyacrylamide gel, electrophoresed, and transferred to PVDF. The total protein was detected by staining with PonceauS (P7170, MilliporeSigma) prior to scanning. The membranes were blocked with a blocking solution (5% nonfat milk in 0.1% Tween-20 in TBS) for 1 h, incubated with primary antibodies diluted in blocking solution overnight at 4 °C, washed four times × 5 min with 0.1% Tween-20 in TBS (TBST), incubated with fluorescence-based secondary antibodies (LI-COR Biosciences) in blocking solution for 1 h, then washed again four times in TBST. The membrane was then dried for 1 h at room temperature prior to being imaged using an Odyssey CLx imager (LI-COR Biosciences). Antibodies used for immunoblotting in this study included: Streptavidin (926-68079, LI-COR Biosciences), LMNB1 (AB16048, Abcam), FLAG (14793, Cell Signaling Technology), HA (901501, BioLegend), V5 (R960-25, Thermo Fisher), TP53 (sc-126, Santa Cruz), GM130 (610822, BD Biosciences) and GAPDH (G9545-100UL, MilliporeSigma).

### Fluorescence staining, fixed imaging, and live-cell Imaging

For fixed cell imaging, cells were grown as indicated on No. 1.5 coverslips, washed once with PBS, and fixed with either 4% PFA or −20 °C methanol for 10 min. The subsequent steps were performed at room temperature unless otherwise denoted. The coverslips were then rinsed with 0.1% Triton X-100 in PBS (PBST) 3x, then permeabilized (in the case of PFA fixation) with 0.3% PBST for 20 min. 3x more washes in 0.1% PBST were carried out prior to blocking with 5% BSA in PBST for 30 min. If the experiment called for primary antibodies, samples were incubated with primary antibodies diluted in a blocking solution for 1 h, then washed 3x with 0.1% PBST. Complementary secondary antibodies (Alexa Secondaries, Thermo Fisher), combined with any cellular stains such as DAPI, were incubated for 1 h. Coverslips were washed 3x 0.1% PBST and mounted on slides using Prolong Gold (P36930, Thermo Fisher). Antibodies used for immunofluorescence in this study included Previously listed antibodies, ARL13B (sc-515784, Santa Cruz), and CEP135 (produced in-house). For the selectable knock-out experiments, Nutlin-3a (18585, Cayman Chemical) was used at 10 µM to induce TP53 accumulation in cells for detection.

For fixed imaging, spinning-disk confocal microscopy was performed using either a Nikon W1-CSU-SoRa or a Nikon CrestOptics-X-Light-DeepSIM microscope equipped with a CFI Plan Apo IR 60XC WI water immersion objective (NA 1.27). Each field was acquired where the whole cell volume of 20 µm could be captured with a z-step of 0.3 µm. Displayed are maximum-intensity projections. For live imaging, 100,000 cells were seeded per well in a four-well Lab-Tek II chamber slide. To capture live-cell snapshots, the cells were treated with SiR live-cell dyes targeting actin, tubulin, or DNA (Spirochrome) at a concentration of 100 nM for 2 h before imaging. Cells were maintained at 37 °C and 5% $CO_2$ using a Tokai Hit environmental control chamber. For the photobleaching assay, RPE-1 PAC (−) cells endogenously tagged to express ACTB-mNeon or ACTB-mStayGold were incubated with 100 nM SiR-DNA for 2 h prior to imaging using a Nikon CrestOptics-X-Light V2 spinning-disk microscope outfitted with a CFI Plan Apo IR 60XC WI water immersion objective (NA 1.27). For each field of view, samples were imaged every min for 20 min at 0.5 µm Z-step size for 15 µm total, first using the 635 nm laser to image SiR-DNA

followed by a 470 nm laser to image the fluorescently tagged proteins. The 470 nm laser was left on between time points to continuously illuminate the sample. Finally, volumetric live-cell super-resolution microscopy was performed on the Nikon CrestOptics-X-Light-DeepSIM microscope in SIM mode. To capture mStayGold-ACTB dynamics, 17 structured illumination images were captured at each z-step of 0.3 µm totaling 6 µm axially. Imaging occurred every 5 min for a total of 4 h. SIM reconstruction was then carried out to achieve super-resolution. Displayed are maximum-intensity projections.

### Software and analysis

General image processing was carried out with Nikon NIS-elements or ImageJ (Schindelin et al, 2012) for deconvolution, maximum-intensity projections, contrast modifications, and counting quantifications. For the photobleaching quantification, unprocessed, maximum-intensity projections were analyzed using the following pipeline. Nuclear masks were obtained using the Stardist (Schmidt et al, 2018) plug-in in ImageJ. The background was subtracted from the DNA channel using a 100-pixel rolling ball method, resized to 60% and analyzed using the default parameters in Stardist. Nuclear masks were imported into Cellprofiler (Stirling et al, 2021) with the ACTB-fusion channel images. Nuclei were filtered using a size cut off to exclude spurious objects, and final nuclear objects were used as a seeding point to detect the cell boundaries of the ACTB-fusion channel using the Propagation method. The background of the ACTB-fusion channel was estimated using the lower quartile of the entire image and subtracted from the channel. The corrected ACTB-fusion channel was masked using the cell boundary objects and the total intensities were measured. The average object total intensity per image was plotted. Four fields of views were analyzed for each biological replicate ($n = 3$). General DNA sequence viewing, annotation, primer design, and cloning design was carried out in SnapGene 7.1.2. Flow cytometry analysis and visualization was performed in BD FlowJo 10.10. Samples were gated to exclude debris, and cell doublets and +GFP fluorescence gates were determined from negative, unedited lines as well as auto-fluorescent controls. Data analysis and visualization were performed in GraphPad Prism 10.2.

### Graphics

Schematics and diagrams were created in part by BioRender.com and further modified in Adobe Illustrator (Synopsis Graphic, Figs. 1, 2C, 3B,F, 4B, 5A and EV1h).

## Data availability

This study includes no data deposited in external repositories.

The source data of this paper are collected in the following database record: biostudies:S-SCDT-10_1038-S44318-024-00337-5.

## Peer review information

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

## Acknowledgements

We thank members of the Pelletier Lab for their scientific feedback during the project. Studentships and Fellowship funding this work include Lunenfeld-Tanenbaum Research Institute Studentships at Sinai Health System to RP. A Hold'em For Life Oncology Fellowship to AS and CIHR Post Doctoral Grants (#187836 and #181763, to AS and ACE, respectively). This project is also funded by a CIHR Project Grant to HDMW (#156297), a CIHR Foundation grant (FDN #167279), and a Krembil Foundation grant to LP. LP is a Tier 1 Canada Research Chair in Centrosome Biogenesis and Function. This work was supported by the Lunenfeld-Tanenbaum Research Institute Flow Cytometry Core and The Network Biology Collaborative Centre, which are funded by the Canada Foundation for Innovation, the Ontario Government, and Genome Canada and Ontario Genomics (OGI-139) and the Nikon Center of Excellence at the Lunenfeld-Tanenbaum Research Institute.

## Author contributions

**Reuben Philip**: Conceptualization; Data curation; Formal analysis; Validation; Investigation; Visualization; Methodology; Writing—original draft; Writing—review and editing. **Amit Sharma**: Conceptualization; Data curation; Formal analysis; Validation; Investigation; Visualization; Methodology; Writing—review and editing. **Laura Matellan**: Data curation; Validation; Investigation; Methodology; Writing—review and editing. **Anna C Erpf**: Data curation; Validation; Investigation; Methodology; Writing—review and editing. **Wen-Hsin Hsu**: Data curation; Validation; Investigation; Methodology; Writing—review and editing. **Johnny M Tkach**: Conceptualization; Data curation; Formal

analysis; Validation; Investigation; Visualization; Methodology; Writing—
review and editing. **Haley D M Wyatt**: Resources; Writing—review and editing.
**Laurence Pelletier**: Resources; Supervision; Funding acquisition; Investigation;
Writing—original draft; Project administration; Writing—review and editing.

Source data underlying figure panels in this paper may have individual
authorship assigned. Where available, figure panel/source data authorship is
listed in the following database record: biostudies:S-SCDT-10_1038-S44318-
024-00337-5.

## Disclosure and competing interests statement
The authors declare no competing interests.

# Expanded View Figures

**Figure EV1. Editing, enrichment, and validation of fluorescent knock-ins with qTAG cassettes across different human cell lines using various mammalian selections.** ▶

(A–C) Left—Representative images of WT, HDR, and MMEJ-targeted H2BC11-moxGFP-Puro tagging in HAP1, ARPE-19, and U-2 OS cells co-stained with DAPI. Scale bars: 10 µm. Right—Flow cytometry quantifications of GFP-positive cells, based on three distinct biological replicates with each measurement encompassing 200,000 cells (Right). Data were mean ± SEM. (D) Genomic PCR outside the homology arms probing for locus-specific integration of the qTAG cassette in HAP1, ARPE-19, and U-2 OS cells. (E) Left—Representative images of WT, selected H2BC11-moxGFP-Blast, selected H2BC11-moxGFP-Puro, and selected H2BC11-moxGFP-Zeo HEK293T cells co-stained with DAPI (Left). Scale bars: 10 µm. Right—Flow cytometry quantifications of GFP-positive cells, based on three distinct biological replicates with each measurement encompassing 200,000 cells (Right). Data were mean ± SEM. (F) Genomic PCR outside the homology arms probing for locus-specific integration of the qTAG cassettes with alternative mammalian selectable markers in HEK293T cells. (G) Overview of an alternate strategy using electroporation and RNPs to edit the tubulin TUBA1B gene with a qTAG-Blast-mScarlet cassette in H9 stem cells. (H) Representative images of WT and Blast-mScarlet-TUBA1B H9 cells co-stained with DAPI and probed for pluripotency marker OCT4. Scale bar: 10 µm. Source data are available online for this figure.

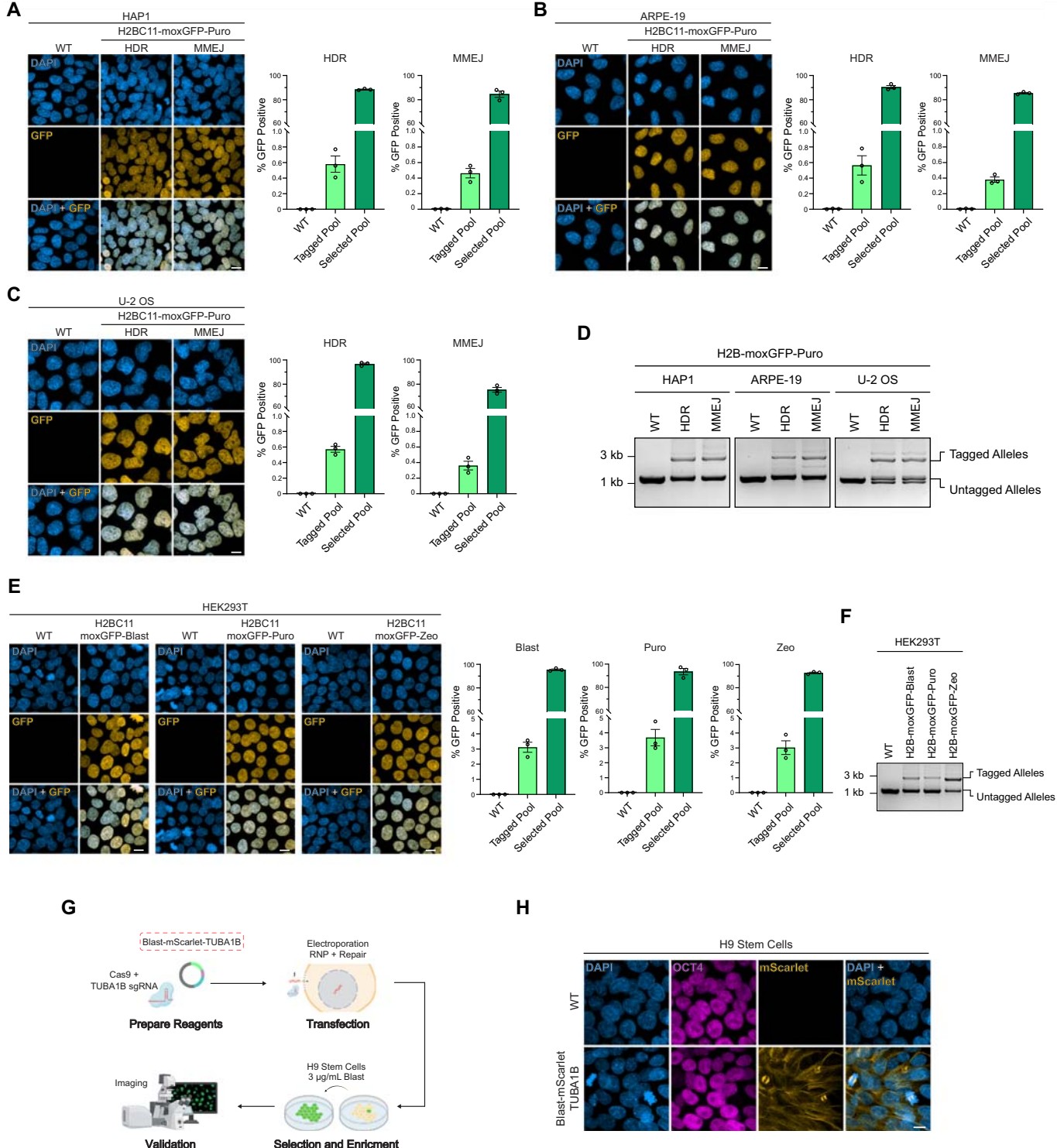

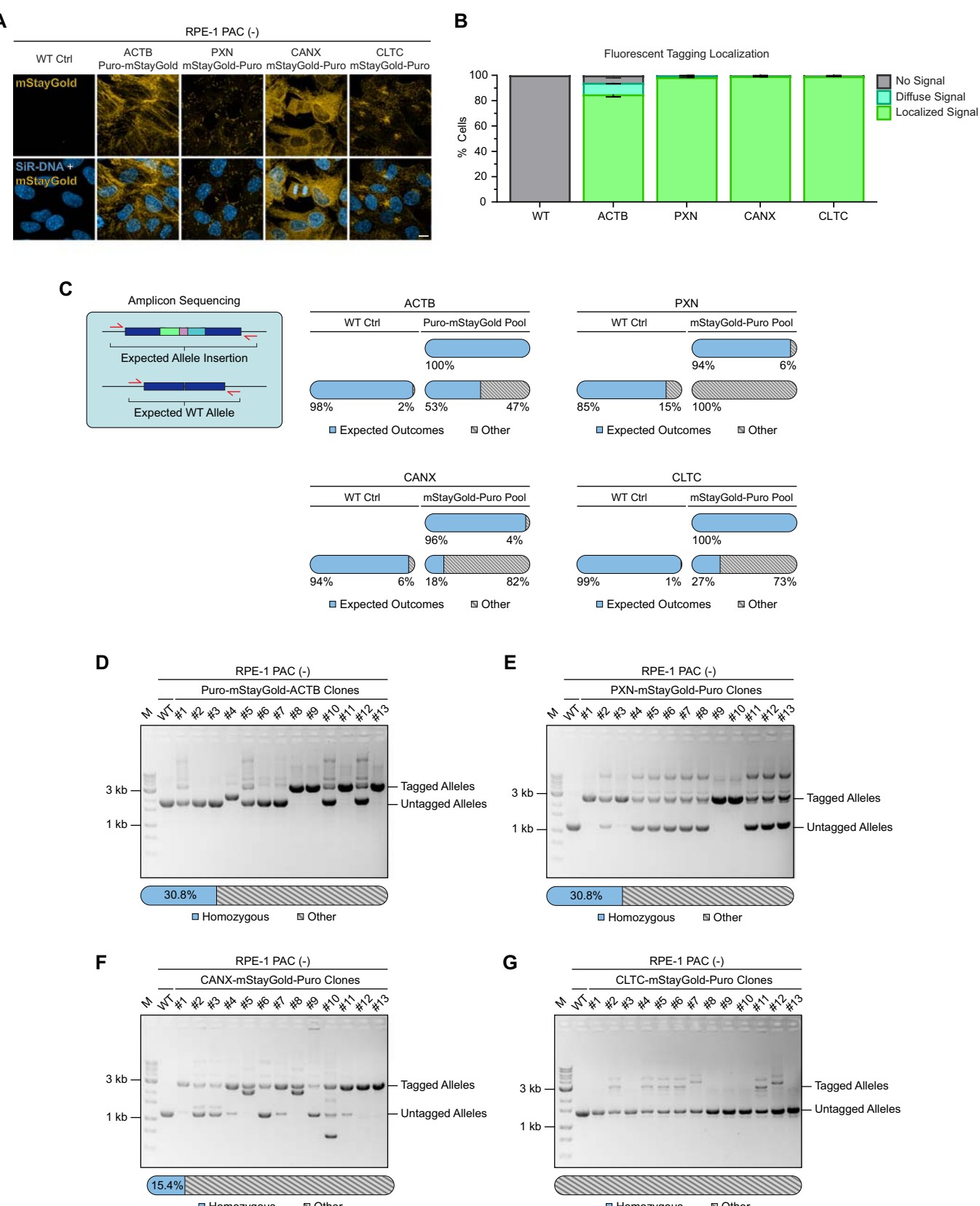

◀  **Figure EV2.  qTAG localization efficiency and editing outcomes using fluorescent cassettes.**

(A) Representative images of WT, and fluorescently tagged-selected pools of various genes, including ACTB, PXN, CANX, and CLTC co-stained with SiR-DNA. Scale bar: 10 μm. (B) Quantification of cells expressing no, diffuse, or localized endogenous fusion signal in antibiotic-selected pools of cells. >500 cells were quantified in each of the three biological replicates for each condition. Data were mean ± SEM. (C) Amplicon sequencing results from dominant amplicons in WT and selected edited cell pools. Expected edited outcomes are highlighted in blue, while gray represents the fraction of mutated sequences. (D–G) Genomic PCR targeting regions outside the homology arms to assess locus-specific integration of the qTAG cassette, serving as an indicator of clonal efficiency. Homozygous cells are highlighted in blue, while the proportion of other outcomes are shown in gray. Source data are available online for this figure.

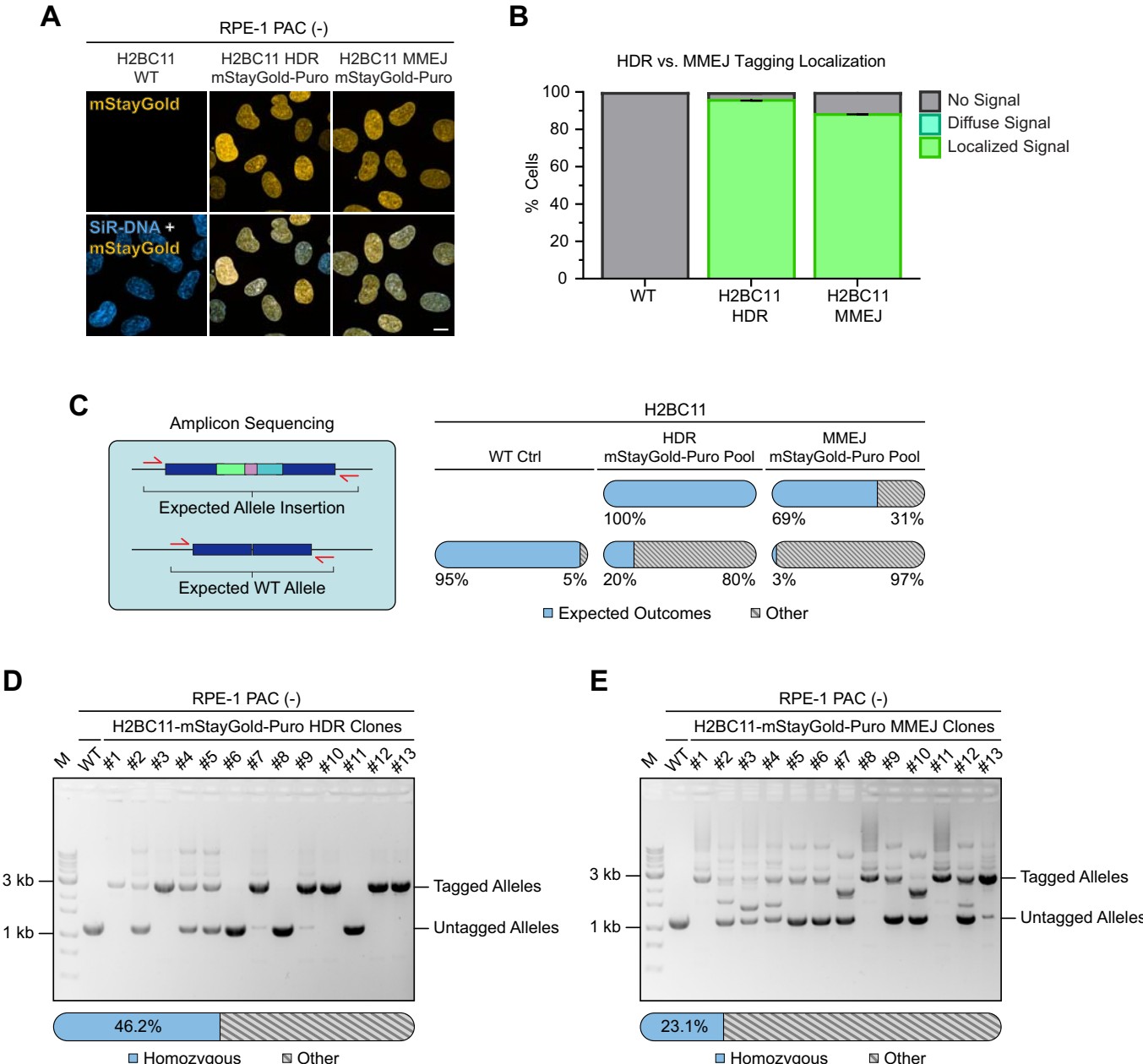

**Figure EV3. qTAG localization efficiency and editing outcomes using cassettes targeting the HDR pathway vs MMEJ pathway.**

(**A**) Representative images of WT, and fluorescently tagged-selected pools H2BC11-mStayGold edited with cassettes targeting the HDR pathway or MMEJ pathway. Scale bars: 10 μm. (**B**) Quantification of cells expressing no, diffuse, or localized endogenous fusion signal in antibiotic-selected pools of cells. >500 cells were quantified in each of the three biological replicates for each condition. Data were mean ± SEM. (**C**) Amplicon sequencing results from dominant amplicons in WT and selected edited cell pools. Expected edited outcomes are highlighted in blue, while gray represents the fraction of mutated sequences. (**D**, **E**) Genomic PCR targeting regions outside the homology arms to assess locus-specific integration of the qTAG cassette, serving as an indicator of clonal efficiency. Homozygous cells are highlighted in blue, while the proportion of other outcomes are shown in gray. Source data are available online for this figure.

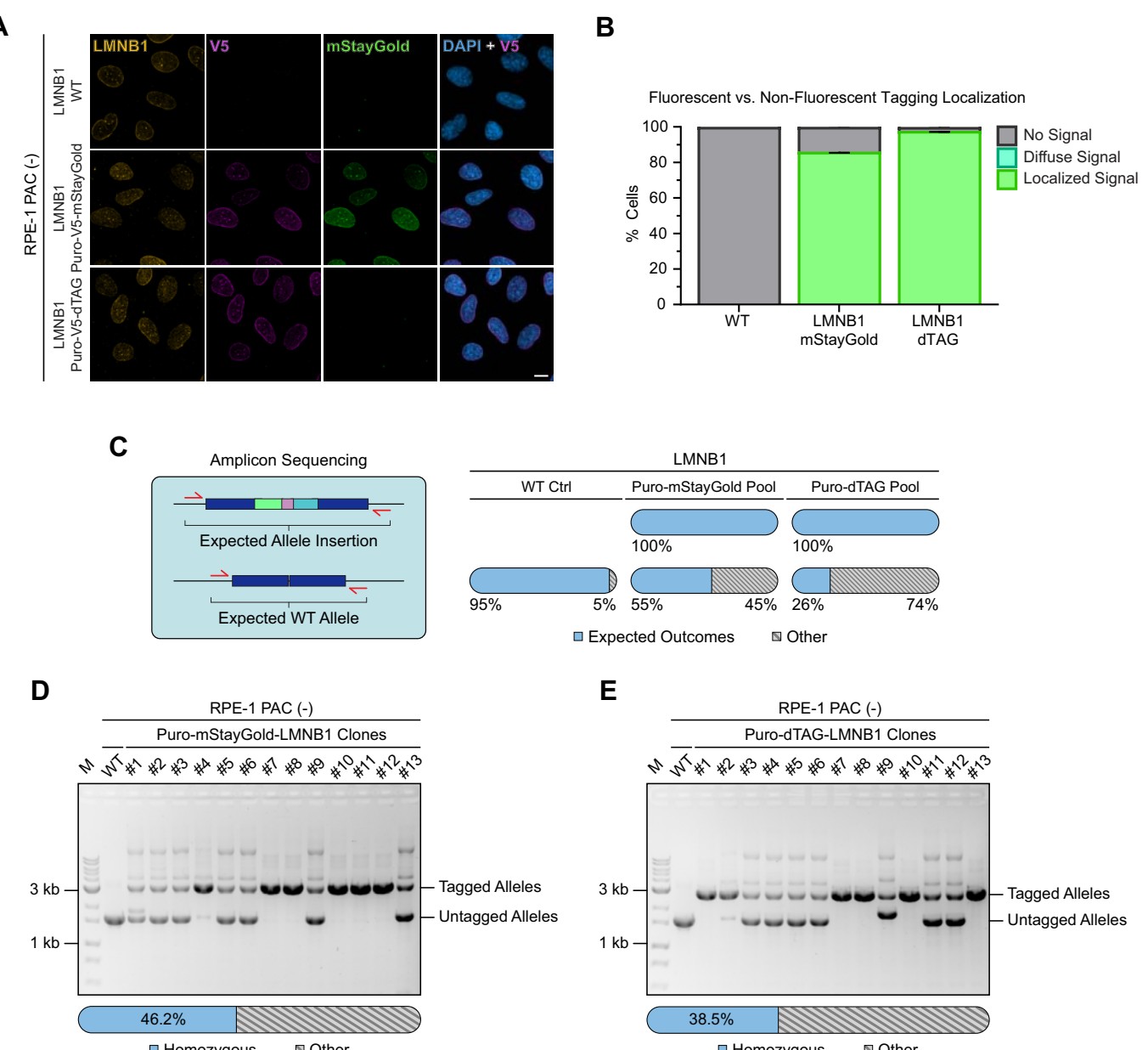

**Figure EV4. qTAG localization efficiency and editing outcomes using fluorescent tagging cassettes vs. non-fluorescent tagging cassettes.**

(A) Representative images of WT, fluorescently tagged-selected pools of mStayGold-LMNB1, and non-fluorescently tagged-selected pools of dTAG-LMNB1. Scale bar: 10 μm. (B) Quantification of cells expressing no, diffuse, or localized endogenous fusion signal in antibiotic-selected pools of cells. >500 cells were quantified in each of the three biological replicates for each condition. Data were mean ± SEM. (C) Amplicon sequencing results from dominant amplicons in WT and selected edited cell pools. Expected edited outcomes are highlighted in blue, while gray represents the fraction of mutated sequences. (D, E) Genomic PCR targeting regions outside the homology arms to assess locus-specific integration of the qTAG cassette, serving as an indicator of clonal efficiency. Homozygous cells are highlighted in blue, while the proportion of other outcomes are shown in gray. Source data are available online for this figure.

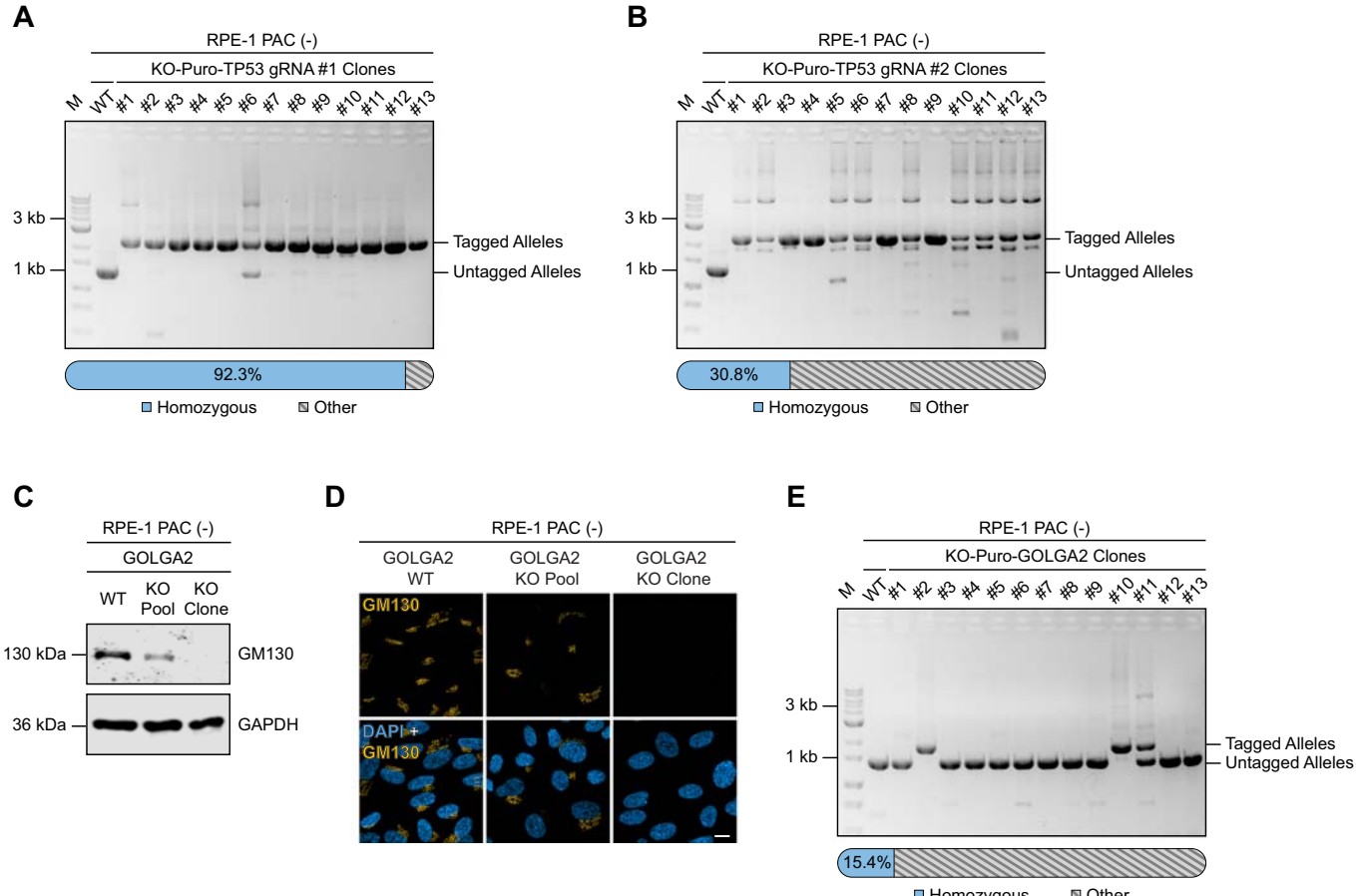

**Figure EV5. Knockout tagging clonal efficiency.**

(A, B) Genomic PCR targeting regions outside the homology arms of TP53 to assess locus-specific integration of the qTAG cassette, serving as an indicator of clonal efficiency. Homozygous cells are highlighted in blue, while the proportion of other outcomes are shown in gray. (C) A representative immunoblot of RPE-1 PAC (−) WT-GOLGA2 cells, a GOLGA2-KO-Puro selected pool of cells and a GOLGA2-KO-Puro selected clonal cell line probed with antibodies against GM130 and GAPDH. (D) Representative images of WT-GOLGA2 cells, a GOLGA2-KO-Puro selected pool of cells, and a GOLGA2-KO-Puro selected clonal cell line probed with an antibody against GM130 and co-stained with DAPI. Scale bar: 10 μm. (E) Genomic PCR targeting regions outside the homology arms of GOLGA2 to assess locus-specific integration of the qTAG cassette. Homozygous cells are highlighted in blue, while the proportion of other outcomes are shown in gray. Source data are available online for this figure.

