## [Peer Review File · The EMBO Journal]

qTAG: An adaptable plasmid scaffold for CRISPR-based endogenous tagging

Reuben Philip, Amit Sharma, Laura Matellan, Anna Erpf, Wen-Hsin Hsu, Johnny Tkach, Haley Wyatt, and Laurence Pelletier

Corresponding author(s): Laurence Pelletier (pelletier@lunenfeld.ca)

Review Timeline:

Transfer from Review Commons:	30th Sep 24
Editorial Decision:	22nd Oct 24
Revision Received:	12th Nov 24
Accepted:	2nd Dec 24

Review
COMMONS

Editor: Hartmut Vodermaier

Transaction Report: This manuscript was transferred to The EMBO JOURNAL following peer review at Review Commons.

Review #1

1. Evidence, reproducibility and clarity:

Evidence, reproducibility and clarity (Required)

The manuscript entitled "A versatile plasmid scaffold for CRISPR-based endogenous tagging" by Philip and colleagues presents a toolkit for endogenous tagging of genes in human cell lines.

Tagging of endogenous genes combines the advantages of optimized epitope tags, which can be used for the detection and purification of proteins with high affinity and specificity, with the opportunities of genome editing preserving a transcripts transcriptional and potentially also post-transcriptional regulation.

The authors considered different categories of epitope tags that are optimized for fluorescent visualization, biochemical purification, proximity ligation, and protein degradation for N- and C-terminal tagging and present a strategy to add any other tag.

Endogenous tagging requires the generation of an in-frame donor plasmid with appropriate homology arms. The authors generated a set of donor plasmids with convenient restriction site for insertion of appropriate homology arms. Their versatile design enables rapid generation of donor plasmids for different genes.

Here, the authors developed a comprehensive toolkit for rapid tagging of endogenous genes in cell lines. The manuscript explains different strategies and alternative options for low-expressed genes and present examples of successfully edited genes in different cell lines.

Conclusion

Overall, the authors present a comprehensive toolkit that combines the advantages of genome editing and epitope-tagging for tissue culture experiments. The manuscript is well-structured and written, and the Figures clearly illustrate strategies and results. I have no doubt that this manuscript and the associated library of plasmids will be useful to a broad community.

Minor comments:

To further facilitate use by others, the authors could add some more discussion on best practice to test the functionality of edited alleles. Assessment of protein levels, subcellular localization, and activity are essential to ensure the functionality of the endogenously tagged protein. While T2A is when tolerated by some mRNAs, it can negatively impact the production of the tagged protein for others. The authors could discuss this in more detail when introducing their elegant addition of loxP sites for post-selection excision of the marker. The authors could also discuss an alternative strategy that uses intronic space to accommodate a selection marker with only minor changes to the resulting mRNA (PMID: 35639929).

I applaud the authors' effort to share this wonderful toolkit with the research community through the addgene repository!

2. Significance:

Significance (Required)

Adding high-affinity epitope tags to endogenous genes has become the novel state-of-the art to interrogate the function of individual genes. However, most current strategies to tag rely on clonal selection of successfully edited cells and expansion of cell clones. This process is laborious, time consuming, and expensive. It also requires that the edited cells can be expanded clonally. Many non-cancerous or primary cells that do not support clonal growth are not amenable to these protocols. To overcome this obstacle, the authors added a selection maker to their donor cassette that enables rapid selection of successfully edited cell pools. The selection marker is either connected to the tagged gene by a T2A peptide, enabling the production of independent polypeptides from the same mRNA by inducing ribosomal skipping, or driven by an independent promoter. Constructs with independent promoters are required to select for endogenous genes that are not or only lowly expressed. Both options are considered by the authors.

The genomic insertion of selection markers greatly facilitates the selection of successfully edited cell pools, avoids unwanted clonal effects, and enables editing of cells that can't be clonally expanded. However, this strategy adds unwanted sequences to the target gene's mRNAs or genomic region and disconnects potential regulation by 5' and 3' untranslated regions. The authors present a solution for this problem by flanking their selection markers with lox sites that enable precise excision of the selection cassette after stable cell pools are generated.

The presented comprehensive toolkit will appeal to a broad audience and facilitate research in ex vivo cell culture systems.

3. How much time do you estimate the authors will need to complete the suggested revisions:

Estimated time to Complete Revisions (Required)

(Decision Recommendation)

Less than 1 month

4. Review Commons values the work of reviewers and encourages them to get credit for their work. Select 'Yes' below to register your reviewing activity at Web of Science Reviewer Recognition Service (formerly Publons); note that the content of your review will not be visible on Web of Science.

No

Review #2

1. Evidence, reproducibility and clarity:

Evidence, reproducibility and clarity (Required)

In Philip et al. the authors have generated a plasmid system to edit genes at their genomic locus using crispr cas9 and demonstrated their usage in mammalian cells in tissue culture. They were able to tag a wide variety of genes with different tags in multiple cell lines and improved this process by conintroduction of antibiotic resistance with 2A or Efs, which can also be removed later on for reuse. Further more they demonstrated usage of specific tags enabling approaches such as controlled targeted degradation, proximity labeling and super resolution imaging. Lastly, they also modified their vector system to improve knockout efficiency or safe-harbor insertion of transgenes.

****Major comments****

Overall, the approaches were well presented, experiments were thoughtfully chosen and beautifully illustrated in particular the tagging related sections. The efficacy of their approach was strongly supported by the data presented. I do not see a necessity to add any experiments.

****Minor comments:****

The drug selection for target knock-out approach could benefit from quantification of the fraction of cells that were p53- to be more convincing (Fig6 H). I was surprised there were no heterozygotes within the population. OPTIONAL: Show improvement compared to sgRNA only and test additional targets.

For the safe harbor system, (Fig6 K) quantification would be useful. The authors could also comment on other existing cas9 safe harbor systems. (e.g. Ocegüera-Yanez et al. 2016). OPTIONAL: Efficacy comparison with existing vector systems.

****Referee Cross-commenting****

I believe all the comments provided by other reviewers are well justified and would help the paper to improve.

2. Significance:

Significance (Required)

Philip et al. combined the currently existing crispr/cas9 knock in approaches with a wide array of tags and resistance markers. Even though they are not the first to report on this technology, the methods are written with enough details to be useful for someone who wishes to carry out endogenous tagging for the first time with tissue culture experience only. The authors made big and impressive effort to make the vast library of tagging and resistance cassettes available to the community via addgene. I believe this paper will be useful for many scientists working with tissue culture and therefore support its publication.

3. How much time do you estimate the authors will need to complete the suggested revisions:

Estimated time to Complete Revisions (Required)

(Decision Recommendation)

Cannot tell / Not applicable

4. Review Commons values the work of reviewers and encourages them to get credit for their work. Select 'Yes' below to register your reviewing activity at Web of Science Reviewer Recognition Service (formerly Publons); note that the content of your review will not be visible on Web of Science.

No

Review #3

1. Evidence, reproducibility and clarity:

Evidence, reproducibility and clarity (Required)

Philip et al., present in their study titled "A versatile plasmid scaffold for CRISPR-based endogenous tagging," the development of a versatile collection of repair cassettes in the form of plasmids, which they name the qTAG system, for CRISPR-mediated endogenous gene tagging in mammalian cells. The authors have designed cassettes for tagging both the N- and C-terminus of the gene-encoded proteins. Additionally, they have integrated standardized cloning sequences and unique restriction sites to enhance the simplicity and adaptability of the system.

Their methodology includes features for serial gene-tagging by flanking the selection marker with loxP sites and employing transient Cre recombinase to induce marker loss, a strategy that is frequently used in yeast. The versatility of this system is showcased through various applications, including fluorescence imaging, proximity-dependent biotinylation, epitope tagging, and targeted protein degradation.

While the manuscript presents a well-structured methodology and results, it lacks a comprehensive evaluation of tagging efficiency and a comparative analysis of the tagging efficiency with existing tagging systems. Furthermore, the study does not offer substantial novelty but serves as a protocol and resource paper.

****Major Points:****

1. In figure 2c-2e, the authors perform imaging and flow cytometry to validate the target integration. However, GFP-positive does not necessarily indicate correct integration of the target. On-target integration with small indels or by NHEJ instead of HDR could also result in a positive GFP signal at

the correct localization, but such tagging-differences may occasionally lead to differences in protein behavior due to the duplication of the sequence region of the homology arm. Also, off-target integration may occasionally lead to the expression of the tag. Among the GFP+ cells, how many have correct integrations with base-pair precision? The authors could (illumine-) sequence insertion junctions such as the band in figure 2f to check for mutations?

2. Did they observe any GFP signal in unexpected cellular location? They need to report the appearance and frequency of the by-products.

3. The authors arbitrarily categorize the repairing mechanisms of their qTAG system into HDR and MMEJ based solely on the homology arm length. The double-strand break can be repaired through multiple pathways. Even with long homology arm, NHEJ-directed repair could still happen (PMID: 34672952). It is essential to sequence the target locus before claiming HDR- or MMEJ-mediated editing has taken place.

4. In non-fluorescent tagging applications, the authors should describe the efficiency of the tagging method. For example, what fraction of the clones have bi-allelic integration with the tag? Compared to the star protocol published by Damhofer et al. (PMID: 34151298), does their qTAG system show any improvement in terms of efficiency or novelty, given that the procedures are quite similar?

5. The authors claimed that "current tagging strategies are typically not easily amenable to switching tags or selectable markers and involve intricate cloning steps", making them less attractive for many labs to adopt". However, in PMID: 32406907a cloning-free method is described that uses simple PCR-generated donors for HDR. This seems to work quite well as shown for a large range of genes, and to enable even simultaneous double-tagging of two genes. To cite this study in the context of a sentence that claims that existing methods are characterized by "limitations [that] often result in extremely low integration rates" seems to be not appropriate. The advantages and disadvantages of their method and other methods, in particular this PCR-based tagging method, should be discussed thoroughly and as fair as possible so that potential users can choose the method that suits them best.

****Minor comments:****

1. On page 8, "Tas an extension of this application" should be corrected to "as an extension of this application".

2. Significance:

Significance (Required)

Endogenous gene tagging facilitates the study of proteins in their native expression context, minimizing the likelihood of artifacts resulting from overexpression. The advent of CRISPR/Cas technology has significantly enhanced the precision and efficiency of endogenous tagging. With a diverse range of available strategies and tags, endogenous tagging finds applications across numerous biological disciplines, from basic research to clinical applications.

In their manuscript, the authors develop a collection of plasmids to serve as repair templates for CRISPR-mediated endogenous tagging. They incorporated features to enhance the simplicity and

adaptability of the system, providing versatile examples to demonstrate the application of different types of tags. However, their work lacks novelty compared to existing tagging systems. For instance, the CRISPR-Cas12a-assisted PCR tagging system (PMID: 32406907) offers options for different fluorescent proteins and selection markers. Similarly, for targeted protein degradation, the published STAR protocols (PMID: 34151298; 34849487) employ comparable procedures.

Additionally, the authors did not demonstrate an improvement or advantage of their system in terms of efficiency. Achieving efficient and precise integration of long genetic fragments without unintended mutations or "scars" or local sequence duplications due to integration by NHEJ remains a significant challenge in endogenous tagging, as it is crucial for maintaining the integrity and functionality of the tagged constructs. Extensive efforts have been made to profile the integration outcomes of DNA double-strand break repair, revealing a surprising degree of repair complexity (PMID: 34672952; 36284361). Unfortunately, this work did not address this critical issue.

This work is well presented, with detailed workflows, methodologies, and clear results, making it potentially valuable as a protocol or a resource paper. The structured presentation enhances its usability for researchers seeking practical guidance in CRISPR-mediated endogenous tagging.

3. How much time do you estimate the authors will need to complete the suggested revisions:

Estimated time to Complete Revisions (Required)

(Decision Recommendation)

Less than 1 month

Yes

We extend our gratitude to the reviewers for their enthusiasm regarding our work, their careful comments, and their many thoughtful suggestions. Below, we provide a detailed point-by-point response, with our replies in **blue**, and the original comments in their entirety presented in *italics*. The referee reports have been immensely helpful, and we hope that the reviewers will find the revisions of our manuscript suitable for publication in EMBO J.

Reviewer #1

Evidence, reproducibility and clarity:

The manuscript entitled "A versatile plasmid scaffold for CRISPR-based endogenous tagging" by Philip and colleagues presents a toolkit for endogenous tagging of genes in human cell lines.

Tagging of endogenous genes combines the advantages of optimized epitope tags, which can be used for the detection and purification of proteins with high affinity and specificity, with the opportunities of genome editing preserving a transcripts transcriptional and potentially also post-transcriptional regulation.

The authors considered different categories of epitope tags that are optimized for fluorescent visualization, biochemical purification, proximity ligation, and protein degradation for N- and C-terminal tagging and present a strategy to add any other tag.

Endogenous tagging requires the generation of an in-frame donor plasmid with appropriate homology arms. The authors generated a set of donor plasmids with convenient restriction site for insertion of appropriate homology arms. Their versatile design enables rapid generation of donor plasmids for different genes.

Here, the authors developed a comprehensive toolkit for rapid tagging of endogenous genes in cell lines. The manuscript explains different strategies and alternative options for low-expressed genes and present examples of successfully edited genes in different cell lines.

Conclusion: Overall, the authors present a comprehensive toolkit that combines the advantages of genome editing and epitope-tagging for tissue culture experiments. The manuscript is well-structured and written, and the Figures clearly illustrate strategies and results. I have no doubt that this manuscript and the associated library of plasmids will be useful to a broad community.

We would like to thank this reviewer for their kind comments. We are glad to hear they share our view that this will be a tremendously useful resource for the broader research community.

Minor comments:

To further facilitate use by others, the authors could add some more discussion on best practice to test the functionality of edited alleles. Assessment of protein levels, subcellular localization, and activity are essential to ensure the functionality of the endogenously tagged protein. While T2A is when tolerated by some mRNAs, it can negatively impact the production of the tagged protein for others. The authors could discuss this in more detail when introducing their elegant addition of loxP sites for post-selection excision

of the marker. The authors could also discuss an alternative strategy that uses intronic space to accommodate a selection marker with only minor changes to the resulting mRNA (PMID: 35639929).

I applaud the authors' effort to share this wonderful toolkit with the research community through the Addgene repository!

Again, we would like to thank this reviewer for their kind comments. They raise three separate points in the above minor comments:

1. *To further facilitate use by others, the authors could add some more discussion on best practice to test the functionality of edited alleles. Assessment of protein levels, subcellular localization, and activity are essential to ensure the functionality of the endogenously tagged protein.*

This is an excellent suggestion. We now include a short discussion on this in the revised version of the manuscript (Lines: 417 – 427):

“Lastly, we present scenarios in which we test various non-fluorescent tags, highlighting tagging situations that are particularly challenging to enrich and isolate without a selection strategy. We validated multiple epitope tags, proximity biotinylation enzymes, and an inducible degron, demonstrating proper localization, verifying endogenous protein levels of the fusions, and assessing the functionality of these fusions. Although we can use methods like genomic PCR, sequencing, immunoblotting, and imaging to confirm the integrity of genetic insertions and the production of the protein fusion, the functionality of the tagged fusions remains uncertain and warrants careful examination. Regardless of whether the fusions are overexpressed or endogenously tagged, they are still considered mutant. Therefore, users should always conduct follow-up phenotypic assays to verify that the protein is functioning as intended.”

2. *While T2A is when tolerated by some mRNAs, it can negatively impact the production of the tagged protein for others. The authors could discuss this in more detail when introducing their elegant addition of loxP sites for post-selection excision of the marker.*

We appreciate the reviewer's insightful comments regarding the impact and potential drawbacks of 2A peptides. We acknowledge that our initial discussion did not adequately emphasize these aspects. In response, we now elaborated further on the potential issues associated with 2A peptides. The details of this have now been included within the revised manuscript (Lines 439-448):

“However, within the standardized structure of the provided qTAG cassettes, two limitations do exist. One drawback is the reliance on transcriptional linkage of the target gene to the selectable marker with the 2A sequence. This requires that the target GOI be expressed at high enough levels in the cell to produce enough antibiotic resistance for efficient selection. Also, while 2A peptide sequences have garnered mass adoption in molecular biology to achieve co-translation, their cleavage or ‘skipping’ efficiency is not always consistent in all cellular backgrounds. This can result in a minor amount of read-through^{1,2} resulting in fusions between the tagged GOI and the selectable marker. The effects on transcript levels, particularly concerning the positioning of the 2A peptide, have also been reported, highlighting the importance of Cre excision in the qTAG system¹.”

3. *The authors could also discuss an alternative strategy that uses intronic space to accommodate a selection marker with only minor changes to the resulting mRNA (PMID: 35639929).*

We appreciate the reviewer for bringing this method to our attention. Initially, we considered conducting a comprehensive analysis of other endogenous tagging systems and summarizing them in a dedicated table to highlight their specific advantages and disadvantages. However, we felt this approach was more appropriate for a review paper. Instead, we opted for a more concise paragraph discussing these methods, organized by their general strategies. Our revised discussion now includes a section that addresses alternative strategies, along with their benefits and limitations, including details on intronic tagging (Lines 383-391):

“Intronic editing systems such as CRISPIE³ face limitations in their applicability. These systems are most suitable for genes with short or single amino acid-encoding exons near their 5' or 3' termini. However, such genes are rare exceptions, restricting the overall utility of intronic tagging approaches. When attempting to target insertions between larger exons, these systems risk creating internal fusions that disrupt the native amino acid sequence and functional proteomic domains. An adaptation of this approach involved inserting an intron tagging cassette⁴ at the N-terminus, offering a clever way to incorporate a promoter-based selection strategy without risking the disruption of the target gene's transcription or the creation of unwanted internal fusion scenarios.”

Significance:

Adding high-affinity epitope tags to endogenous genes has become the novel state-of-the art to interrogate the function of individual genes. However, most current strategies to tag rely on clonal selection of successfully edited cells and expansion of cell clones. This process is laborious, time consuming, and expensive. It also requires that the edited cells can be expanded clonally. Many non-cancerous or primary cells that do not support clonal growth are not amendable to these protocols. To overcome this obstacle, the authors added a selection maker to their donor cassette that enables rapid selection of successfully edited cell pools. The selection marker is either connected to the tagged gene by a T2A peptide, enabling the production of independent polypeptides from the same mRNA by inducing ribosomal skipping, or driven by an independent promoter. Constructs with independent promoters are required to select for endogenous genes that are not or only lowly expressed. Both options are considered by the authors.

The genomic insertion of selection markers greatly facilitates the selection of successfully edited cell pools, avoids unwanted clonal effects, and enables editing of cells that can't be clonally expanded. However, this strategy adds unwanted sequences to the target gene's mRNAs or genomic region and disconnects potential regulation by 5' and 3' untranslated regions. The authors present a solution for this problem by flanking their selection markers with lox sites that enable precise excision of the selection cassette after stable cell pools are generated.

The presented comprehensive toolkit will appeal to a broad audience and facilitate research in ex vivo cell culture systems.

We thank this reviewer for their support and their excitement about our manuscript and the qTAG system.

Reviewer #2

Evidence, reproducibility and clarity:

In Philip et al. the authors have generated a plasmid system to edit genes at their genomic locus using crispr cas9 and demonstrated their usage in mammalian cells in tissue culture. They were able to tag a wide variety of genes with different tags in multiple cell lines and improved this process by co-introduction of antibiotic resistance with 2A or Efs, which can also be removed later on for reuse. Furthermore, they demonstrated usage of specific tags enabling approaches such as controlled targeted degradation, proximity labeling and super resolution imaging. Lastly, they also modified their vector system to improve knockout efficiency or safe-harbor insertion of transgenes.

Major comments:

Overall, the approaches were well presented, experiments were thoughtfully chosen and beautifully illustrated in particular the tagging related sections. The efficacy of their approach was strongly supported by the data presented. I do not see a necessity to add any experiments.

We thank this reviewer for their very positive feedback.

Minor comment:

The drug selection for target knock-out approach could benefit from quantification of the fraction of cells that were p53- to be more convincing (Fig6 H). I was surprised there were no heterozygotes within the population. OPTIONAL: Show improvement compared to sgRNA only and test additional targets. For the safe harbor system, (Fig6 K) quantification would be useful. The authors could also comment on other existing cas9 safe harbor systems. (e.g. Oceguera-Yanez et al. 2016). OPTIONAL: Efficacy comparison with existing vector systems.

The reviewer raises two separate points here:

1. *The drug selection for target knock-out approach could benefit from quantification of the fraction of cells that were p53- to be more convincing (Fig6 H). I was surprised there were no heterozygotes within the population. OPTIONAL: Show improvement compared to sgRNA only and test additional targets.*

We appreciate the reviewer's suggestion regarding the efficiency of the specific sgRNA for the TP53 cassette knock-out. To demonstrate the effectiveness of this KO cassette and ascertain whether this scenario was aided by an exceptional sgRNA, we carried out gene editing, selection enrichment, and the isolation of clones in RPE-1 PAC (-) cells using two distinct sgRNAs against TP53 as well as targeted an additional gene for this KO cassette strategy (Fig. 6f- h., Supp. Fig. 7a-e, Lines 323- 345). By analyzing the clones through PCR, we confirmed that TP53 sgRNA #1 indeed resulted in an unusually high rate of homozygous integrations, likely due to the inherent efficiency of the sgRNA itself. While testing sgRNA #2 did not produce similarly high homozygous integration rates, effective knockout of TP53 within the pool was still achieved. We suspect that sgRNAs targeting the N-terminus encoding region of TP53 may be particularly efficient, and in

combination with our KO cassette, lead to highly effective protein ablation in this case. However, when testing a different gene, GOLGA2, our results showed a modest reduction in protein levels in the pool but still achieved complete ablation in homozygous edited cell lines.

2. *For the safe harbor system, (Fig6 K) quantification would be useful.*

We assume the reviewer is inquiring about the proportion of cells that are GFP positive due to the integration of the safe-harbor GFP-expressing cassette. To assess the localization efficiency of our modified AAVS1 plasmids, we quantified the presence or absence of moxGFP expression in RPE-1 PAC (-) selected pools (Fig. 6l., Lines 358–361). Our findings revealed that over 90% of the cells in within the selected pool exhibited GFP fluorescence in both the PGK- and EF1a-driven AAVS1 donor constructs.

3. *The authors could also comment on other existing cas9 safe harbor systems. (e.g. Ocegüera-Yanez et al. 2016). OPTIONAL: Efficacy comparison with existing vector systems.*

We are grateful to the reviewer for correctly citing other systems that target the AAVS1 locus. Our intention with this figure was to demonstrate the flexibility of our cassette structure for modification, showing its suitability for various tagging scenarios, including exogenous expression at the safe harbor site AAVS1. We did not intend to make direct comparisons between systems. However, the reviewer's point is well taken; the source sequences used to generate the cassette should be cited and discussed. The manuscript has now been amended with these details (Lines 348 – 353):

“The AAVS1 safe harbor site was chosen for the insertion of a modified qTAG cassette to drive gene expression. The design of the homology arms and a portion of the new integration cassette was based on previous strategies used to edit this locus^{5,6}. This cassette removes the core tag-2A-marker structure and replaces it with a selectable marker, a promoter, and a multiple cloning site. This configuration enables the seamless integration of a coding sequence (CDS) of your preference (Fig. 6i).”

Significance:

Philip et al. combined the currently existing crispr/cas9 knock in approaches with a wide array of tags and resistance markers. Even though they are not the first to report on this technology, the methods are written with enough details to be useful for someone who wishes to carry out endogenous tagging for the first time with tissue culture experience only. The authors made big and impressive effort to make the vast library of tagging and resistance cassettes available to the community via addgene. I believe this paper will be useful for many scientists working with tissue culture and therefore support its publication.

We appreciate the reviewer's kind remarks. We are pleased that they recognize the qTAG system's usefulness to scientists and support its publication.

Reviewer #3

Evidence, reproducibility and clarity:

Philip et al., present in their study titled "A versatile plasmid scaffold for CRISPR-based endogenous tagging," the development of a versatile collection of repair cassettes in the form of plasmids, which they name the qTAG system, for CRISPR-mediated endogenous gene tagging in mammalian cells. The authors have designed cassettes for tagging both the N- and C-terminus of the gene-encoded proteins. Additionally, they have integrated standardized cloning sequences and unique restriction sites to enhance the simplicity and adaptability of the system.

Their methodology includes features for serial gene-tagging by flanking the selection marker with loxP sites and employing transient Cre recombinase to induce marker loss, a strategy that is frequently used in yeast. The versatility of this system is showcased through various applications, including fluorescence imaging, proximity-dependent biotinylation, epitope tagging, and targeted protein degradation.

While the manuscript presents a well-structured methodology and results, it lacks a comprehensive evaluation of tagging efficiency and a comparative analysis of the tagging efficiency with existing tagging systems. Furthermore, the study does not offer substantial novelty but serves as a protocol and resource paper.

We would like to thank this reviewer for their thoughtful comments. We are pleased to hear that they find our demonstration of the qTAG system's versatility across multiple tagging scenarios compelling and that they believe it will serve as an important protocol and resource paper, particularly given that the collection will be widely accessible on Addgene. The reviewer raises some general technical issues about endogenous tagging efficiency, repair modalities, and we discuss how our system compares with other tagging methods. We are happy to address these points in the revised manuscript, as detailed below. We want to stress that our primary intention was not to claim superior efficiency compared to other methods, but rather to highlight the versatility, usefulness, and breadth of the qTAG system, a point all three reviewers agree with.

Major points:

1. *In figure 2c-2e, the authors perform imaging and flow cytometry to validate the target integration. However, GFP-positive does not necessarily indicate correct integration of the target. On-target integration with small indels or by NHEJ instead of HDR could also result in a positive GFP signal at the correct localization, but such tagging-differences may occasionally lead to differences in protein behavior due to the duplication of the sequence region of the homology arm. Also, off-target integration may occasionally lead to the expression of the tag. Among the GFP+ cells, how many have correct integrations with base-pair precision? The authors could (illumina-) sequence insertion junctions such as the band in figure 2f to check for mutations?*

While we did provide some metrics of efficiency with our flow cytometry experiments, the reviewer rightly points out that our methods do not confirm whether our edited integrations are precisely in-frame and mutation-free at the base-pair level. To further evaluate the efficiency of this cassette system and assess sequence-level outcomes, we aimed to determine the allelic editing results. After performing editing and selection on a range of genes, we used PCR to

amplify the targeted region and analyzed the amplicons by Nanopore NGS (Supp. Fig. 3c, Supp. Fig. 4c, Supp. Fig. 6c). Amplicons corresponding to the loci with the expected insertions in the edited cell pools consistently showed a high rate of precise editing, while those without the anticipated insertions frequently exhibited moderate to high mutation rates. As the reviewer pointed out, this now offers much clearer insights into the outcomes of the editing process.

2. *Did they observe any GFP signal in unexpected cellular location? They need to report the appearance and frequency of the by-products.*

The reviewer raises a valid point that fluorescent mislocalization could result from mutations or off-target integrations in the genomic sequence, leading to fusion by-products. To address this, we scrutinized a group of edited and selected pools for their target localization by capturing large fields of live cells and quantified the rates of expected vs unexpected localization. We specifically quantified the selected cell pools to determine the rates of no signal localization, diffuse signal localization, or expected localization of the endogenous fusions within the selected pools (Supp. Fig. 3b, Supp. Fig. 4b, Supp. Fig. 6b). Our results indicate that over 80% of the population across all tested genes exhibited localized protein indicative of their organelle or cellular structure, although some genes, such as ACTB, showed a small proportion of cells containing diffuse localization as well as no localization. When comparing our results with contemporary approaches, such as the PCR Tagging method⁷, where similar genes were edited and enriched for using antibiotic selection, we observed different localization rates. In the PCR Tagging method, puro selection for CANX and CLTC resulted in approximately 20% and 40% localized signal, respectively. In contrast, our quantifications for the same genes and antibiotic selection, achieved over 95% localized fluorescent signal in the selected pools for both CANX and CLTC. Additionally, the highest enrichment of localized signal in the PCR Tagging method was observed with POM21 tagging followed by zeocin selection, yielding ~60% localized signal. Our method, however, achieved a minimum of ~80% localization, with many genes showing over 95% localized protein in the selected population. We suspect that this difference in localization rates is likely due to our use of the multicistronic approach rather than the promoter-based method, as selection resistance depends on the successful integration of the cassette in frame with the gene of interest. We appreciate the reviewer's suggestion for this experiment, as it provides a valuable metric that supports the use of our qTAG system.

3. *The authors arbitrarily categorize the repairing mechanisms of their qTAG system into HDR and MMEJ based solely on the homology arm length. The double-strand break can be repaired through multiple pathways. Even with long homology arm, NHEJ-directed repair could still happen (PMID: 34672952). It is essential to sequence the target locus before claiming HDR- or MMEJ-mediated editing has taken place.*

We acknowledge the reviewer's point that even when HDR or MMEJ targeting donor plasmids are transfected into cells, other DNA repair outcomes can still occur. We agree that the presence of a donor template does not guarantee that HDR or MMEJ will be the exclusive repair pathway used in every cell and that other repair pathways and outcomes will occur. We apologize for this imprecise language in our manuscript. To address this, we modified the language throughout the

manuscript to avoid prematurely concluding the DNA repair pathway of the editing process before observing and testing the actual outcomes. An example of this is the rewording of "HDR construct" to "HDR-targeting donor" when referring to qTAG donors with long homology arms.

We were further curious about the differences between targeting either HDR or MMEJ with our qTAG cassettes. Therefore, we compared the tagging of H2BC11 by constructing and performing edits with donors that targeted the HDR and MMEJ repair mechanisms (Supp. Fig. 4a-e). Sequencing revealed that HDR resulted in fewer erroneous integrations within the expected edited allele compared to MMEJ (Lines 172 – 182).

4. *In non-fluorescent tagging applications, the authors should describe the efficiency of the tagging method. For example, what fraction of the clones have bi-allelic integration with the tag? Compared to the star protocol published by Damhofer et al. (PMID: 34151298), does their qTAG system show any improvement in terms of efficiency or novelty, given that the procedures are quite similar?*

We agree with the reviewer's suggestion in describing more efficiency metrics for our assays. In addition to measuring the allelic editing frequency, clonal outcomes were also assessed by selecting clones from the pool to ascertain integration rates at the ploidy level (WT vs Heterozygous vs Homozygous). Clonal allelic outcomes had showed that the rates of homozygous integration varied significantly across the different genes we tested, ranging from 0%- 46% of the clones (Supp. Fig. 3d-g, Supp. Fig. 4d-e, Supp. Fig. 6d-e). In particular, the comparison between fluorescent and non-fluorescent gene tagging when targeting the same locus revealed similar allelic and clonal integration rates, suggesting that there is no substantial difference based on the integrated tag sequence. Unfortunately, we don't believe that comparing rates between systems was very useful in this case as this "efficiency" metric is influenced by numerous variables, such as the cell lines used, Cas9 type and dosage, transfection efficiency, and sgRNA efficacy, which complicates controlled comparisons. We apologize if our manuscript invited any comparison, however, these assays as a result have provided greater clarity and set realistic expectations regarding the performance of the qTAG system.

5. *The authors claimed that "current tagging strategies are typically not easily amenable to switching tags or selectable markers and involve intricate cloning steps", making them less attractive for many labs to adopt". However, in PMID: 32406907a cloning-free method is described that uses simple PCR-generated donors for HDR. This seems to work quite well as shown for a large range of genes, and to enable even simultaneous double-tagging of two genes. To cite this study in the context of a sentence that claims that existing methods are characterized by "limitations [that] often result in extremely low integration rates" seems to be not appropriate. The advantages and disadvantages of their method and other methods, in particular this PCR-based tagging method, should be discussed thoroughly and as fair as possible so that potential users can choose the method that suits them best.*

We apologize for the language used in this section of the manuscript. Our intention was to acknowledge the existence of other systems while presenting our constructs as an alternative

rather than inviting direct comparisons or claiming that one methodology is superior to another. We agree that these alternative strategies, should be discussed and cited in this paper emphasizing their benefits, and limitations to provide users with a comprehensive understanding of available options. For this we added a paragraph within the discussion where these alternative systems are highlighted (Lines 371 – 396). We further removed any language that implies that our system is in any way "better", as direct comparisons of efficiency, mutation rates, and precise editing outcomes are often difficult due to variables like cell types, transfection methods, Cas enzyme delivery, donor delivery method, and cellular selection methods.

“Several different strategies have been developed to address the challenges associated with endogenous tagging in cells^{3,7-15}. These systems incorporate variations like different Cas enzymes, donor repair formats, delivery methods, and post-editing enrichment strategies, but each has its own caveats and limitations. The Allen Institute was among the first to provide a comprehensive set of donor constructs for endogenous tagging in 2017¹⁵. These donors have been instrumental as one of the initial collections designed to label many major cellular organelles. However, a limitation of this implementation is the adoption of a tag-only approach, which lacks positive selection capabilities and relies solely on fluorescent tagging using older, less performant fluorescent proteins and FACS sorting for selection. The PCR tagging system⁷ advocates for simple template production by supplying dsDNA templates in the form of linear PCR cassettes instead of large donor plasmids. These cassettes rely on promoter-based selection and are designed for use with the Cas12 endonuclease. However, this system is limited to C-terminal tagging. Intronic editing systems such as CRISPIE³ face limitations in their applicability. These systems are most suitable for genes with short or single amino acid-encoding exons near their 5' or 3' termini. However, such genes are rare exceptions, restricting the overall utility of intronic tagging approaches. When attempting to target insertions between larger exons, these systems risk creating internal fusions that disrupt the native amino acid sequence and functional proteomic domains. An adaptation of this approach involved inserting an intron tagging cassette⁴ at the N-terminus, offering a clever way to incorporate a promoter-based selection strategy without risking the disruption of the target gene's transcription or the creation of unwanted internal fusion scenarios. More intricate systems such as HIT1¹⁶ editing come with the benefits of reducing undesired NHEJ byproducts via concurrent nicking of the genomic locus in order to integrate the donor target sequence. This approach requires cloning the target sgRNA within the Cas9-nicking construct and embedding sgRNA binding sites in the donor to flank the homology arms, requiring multiple nicking steps to facilitate integration, potentially reducing overall editing efficiencies.”

Minor comments:

1. On page 8, "Tas an extension of this application" should be corrected to "as an extension of this application".

The error within the manuscript was rectified (Line 277).

Significance:

Endogenous gene tagging facilitates the study of proteins in their native expression context, minimizing the likelihood of artifacts resulting from overexpression. The advent of CRISPR/Cas technology has significantly

enhanced the precision and efficiency of endogenous tagging. With a diverse range of available strategies and tags, endogenous tagging finds applications across numerous biological disciplines, from basic research to clinical applications.

In their manuscript, the authors develop a collection of plasmids to serve as repair templates for CRISPR-mediated endogenous tagging. They incorporated features to enhance the simplicity and adaptability of the system, providing versatile examples to demonstrate the application of different types of tags. However, their work lacks novelty compared to existing tagging systems. For instance, the CRISPR-Cas12a-assisted PCR tagging system (PMID: 32406907) offers options for different fluorescent proteins and selection markers. Similarly, for targeted protein degradation, the published STAR protocols (PMID: 34151298; 34849487) employ comparable procedures.

Additionally, the authors did not demonstrate an improvement or advantage of their system in terms of efficiency. Achieving efficient and precise integration of long genetic fragments without unintended mutations or "scars" or local sequence duplications due to integration by NHEJ remains a significant challenge in endogenous tagging, as it is crucial for maintaining the integrity and functionality of the tagged constructs. Extensive efforts have been made to profile the integration outcomes of DNA double-strand break repair, revealing a surprising degree of repair complexity (PMID: 34672952; 36284361). Unfortunately, this work did not address this critical issue.

This work is well presented, with detailed workflows, methodologies, and clear results, making it potentially valuable as a protocol or a resource paper. The structured presentation enhances its usability for researchers seeking practical guidance in CRISPR-mediated endogenous tagging.

The reviewer appreciates the simplicity and adaptability of the qTAG system. We agree that our manuscript is most appropriately categorized as a Protocol/Resource paper, which aligns with our original intent when developing this work. While the qTAG system does not introduce fundamentally new molecular strategies for inserting exogenous sequences, it represents a broad optimization of existing donor construct formats. Its key innovations lie in its user-friendly design, featuring universal cloning adaptors and unique restriction sites flanking each genetic element for easy adaptability. To our knowledge, no other endogenous tagging system is as comprehensive and ready to use in so many applications while also being freely available on Addgene. We believe this is the fundamental strength of the qTAG system and what will encourage greater adoption of endogenous tagging efforts in future studies. We hope that addressing the efficacy and unintended mutations/"scars" issues, as detailed in our point-by-point response, will be satisfactory.

References:

- 1 Liu, Z. *et al.* Systematic comparison of 2A peptides for cloning multi-genes in a polycistronic vector. *Sci Rep* **7**, 2193 (2017). <https://doi.org/10.1038/s41598-017-02460-2>
- 2 Kim, J. H. *et al.* High cleavage efficiency of a 2A peptide derived from porcine teschovirus-1 in human cell lines, zebrafish and mice. *PLoS One* **6**, e18556 (2011). <https://doi.org/10.1371/journal.pone.0018556>
- 3 Zhong, H. *et al.* High-fidelity, efficient, and reversible labeling of endogenous proteins using CRISPR-based designer exon insertion. *Elife* **10** (2021). <https://doi.org/10.7554/eLife.64911>
- 4 Meng, Q. *et al.* Functional editing of endogenous genes through rapid selection of cell pools (Rapid generation of endogenously tagged genes in *Drosophila* ovarian somatic sheath cells). *Nucleic Acids Res* **50**, e90 (2022). <https://doi.org/10.1093/nar/gkac448>
- 5 Oceguera-Yanez, F. *et al.* Engineering the AAVS1 locus for consistent and scalable transgene expression in human iPSCs and their differentiated derivatives. *Methods* **101**, 43-55 (2016). <https://doi.org/10.1016/j.ymeth.2015.12.012>
- 6 Hockemeyer, D. *et al.* Efficient targeting of expressed and silent genes in human ESCs and iPSCs using zinc-finger nucleases. *Nat Biotechnol* **27**, 851-857 (2009). <https://doi.org/10.1038/nbt.1562>
- 7 Fueller, J. *et al.* CRISPR-Cas12a-assisted PCR tagging of mammalian genes. *J Cell Biol* **219** (2020). <https://doi.org/10.1083/jcb.201910210>
- 8 Liang, X., Potter, J., Kumar, S., Ravinder, N. & Chesnut, J. D. Enhanced CRISPR/Cas9-mediated precise genome editing by improved design and delivery of gRNA, Cas9 nuclease, and donor DNA. *J Biotechnol* **241**, 136-146 (2017). <https://doi.org/10.1016/j.jbiotec.2016.11.011>
- 9 Artegiani, B. *et al.* Fast and efficient generation of knock-in human organoids using homology-independent CRISPR-Cas9 precision genome editing. *Nat Cell Biol* **22**, 321-331 (2020). <https://doi.org/10.1038/s41556-020-0472-5>
- 10 Lin, D. W. *et al.* Microhomology-based CRISPR tagging tools for protein tracking, purification, and depletion. *J Biol Chem* **294**, 10877-10885 (2019). <https://doi.org/10.1074/jbc.RA119.008422>
- 11 Sakuma, T., Nakade, S., Sakane, Y., Suzuki, K. T. & Yamamoto, T. MMEJ-assisted gene knock-in using TALENs and CRISPR-Cas9 with the PITCh systems. *Nat Protoc* **11**, 118-133 (2016). <https://doi.org/10.1038/nprot.2015.140>
- 12 Perez-Leal, O. *et al.* Multiplex Gene Tagging with CRISPR-Cas9 for Live-Cell Microscopy and Application to Study the Role of SARS-CoV-2 Proteins in Autophagy, Mitochondrial Dynamics, and Cell Growth. *CRISPR J* **4**, 854-871 (2021). <https://doi.org/10.1089/crispr.2021.0041>
- 13 Cho, N. H. *et al.* OpenCell: Endogenous tagging for the cartography of human cellular organization. *Science* **375**, eabi6983 (2022). <https://doi.org/10.1126/science.abi6983>
- 14 Willems, J. *et al.* ORANGE: A CRISPR/Cas9-based genome editing toolbox for epitope tagging of endogenous proteins in neurons. *PLoS Biol* **18**, e3000665 (2020). <https://doi.org/10.1371/journal.pbio.3000665>
- 15 Roberts, B. *et al.* Systematic gene tagging using CRISPR/Cas9 in human stem cells to illuminate cell organization. *Mol Biol Cell* **28**, 2854-2874 (2017). <https://doi.org/10.1091/mbc.E17-03-0209>
- 16 Bollen, Y. *et al.* Efficient and error-free fluorescent gene tagging in human organoids without double-strand DNA cleavage. *PLoS Biol* **20**, e3001527 (2022). <https://doi.org/10.1371/journal.pbio.3001527>

Dr. Laurence Pelletier
Lunenfeld-Tanenbaum Research Institute
600 University Ave
Toronto, Ontario M5G 1X5
Canada

22nd Oct 2024

Re: EMBOJ-2024-119161-T
An adaptable plasmid scaffold for CRISPR-based endogenous tagging

Dear Laurence,

Thank you again for submitting your revised Review Commons manuscript for consideration by The EMBO Journal. As discussed earlier, I decided to consider the work as a regular EMBO Journal revision, and returned it directly to the original referees 1 and 3. I am happy to say that both consider the manuscript significantly improved and the original concerns well-answered. Following adjustment to our specific journal format and incorporation of a few other editorial modifications (as follows), we should therefore be ready to proceed with acceptance and publication of the study:

GENERAL:

- Please download and complete our author checklist (link provided below).
- Please provide suggestions for a short 'blurb' text prefacing and summing up the conceptual aspect of the study in two sentences (max. 250 characters), followed by 3-5 one-sentence 'bullet points' with brief factual statements of key results of the paper; they will form the basis of an editor-written 'Synopsis' accompanying the online version of the article. Please also upload a synopsis image, which can be used as a "visual title" for the synopsis section of your paper (maybe based on a slimmed-down version of Figure 1?). The image should be in PNG or JPG format, and please make sure that it remains in the modest dimensions of (exactly) 550 pixels wide and 300-600 pixels high.
- You shall also receive a separate message from our Source Data curation team, with instructions on how to prepare and upload relevant image and numerical raw data.

TEXT:

- Please upload the manuscript text as an editable DOCX file, without tracking of earlier changes.
- Please adjust the order of the manuscript sections: Title page with complete author information, Abstract, Keywords, Introduction, Results, Discussion, Methods, Data Availability, Acknowledgements, Disclosure and Competing Interests Statement, References, Main Figure Legends, Tables, Expanded Figure Legends.
- Please remove the Abbreviations glossary on the front page, and instead make sure to define each abbreviation upon first use of the respective term.
- Please provide a more concise abstract, keeping our regular 175 word length in mind. Also, consider making the title somewhat more explicit, e.g. by mentioning the name of your new system already at that point.
- Figure Legends: Please make sure to define the error bars in the legends of figures 2d; 5b; 6e, I; and to appropriately label the axis gaps in figure 2d.
- Please note that Materials and Methods need to be described in the main text using our 'Structured Methods' format (for detail, see <https://www.embopress.org/page/journal/14693178/authorguide#structuredmethods>). The in-text "Methods" section should contain method and protocol descriptions (ideally using a step-by-step protocol format to facilitate adoption of the methodologies across labs), while all key reagents, experimental models, software and relevant equipment - including their sources and relevant identifiers - should be listed in a separately uploaded Reagents and Tools Table, a template for which can be downloaded from the above section of our Author Guidelines.
- As we are switching from a free-text author contribution statement towards a more formal statement based on Contributor Role Taxonomy (CRediT) terms, please remove the present Author Contribution section and instead specify each author's contribution(s) directly in the Author Information page of our submission system during upload of the final manuscript. See <https://casrai.org/credit/> for more information.

- Please rename the Competing Interest section into "Disclosure and Competing Interests Statement", in accordance with our updated Guide to Authors (<https://www.embopress.org/competing-interests>)
- Please adjust the format of the reference list and of the in-text citations according to EMBO Journal format (alphabetical order, author name et al + year, first up to 10 authors should be listed, followed by 'et al' ...).
- Please include a dedicated "Data Availability" section at the end of the Material and Methods (suggested wording: "The [structural coordinates | microarray | mass spectrometry] data from this publication have been deposited to the [name of the database] database [URL] and assigned the identifier [accession | permalink | hashtag]."); should there no data deposition to public repositories linked to the study, this should still be stated as "This study includes no data deposited in external repositories."
- Please make sure to include all relevant funding information not only in the manuscript text, but also in our submission system.

DATA:

- Please refer to our author guide (www.embopress.org/page/journal/14602075/authorguide#expandedview) regarding "supplementary information", and consider re-organizing the current figures and supplemental figures. We are not limited to 6 main figures, and in addition, we can have up to 5 "Expanded View" figures, whose legends would also need to be in the main text, and which would be type-set and directly visible (expandable) with the HTML version of the paper. My suggestion would be to convert 5 of the current supplementary figures into Expanded View figures (naming/in-text callouts: Figure EV1-5), and to promote some other "supplementary" content into additional main figures/panels.
- All main and EV figures should be uploaded as individual files with sufficient resolution/quality for production.
- Please double-check that all microscopy panels have had image data inserted, even in the seemingly blank control panels.
- Please convert the four tables into Expanded View Tables (name/callout: Table EV1-4), moving them out of the text and into individual DOCX or XLSX files including their headers/legends.
- Please convert the "Supplemental Data" into Expanded View Datasets (callout: Dataset EV1-2). Dataset EV1 can be uploaded as ZIP archive with its legend included as readme/legend text file; and Dataset EV2 should be uploaded as XLSX file, with its legend & header included in a separate "legend" tab.

Should you need additional guidance/feedback regarding this final adjustments, please do not hesitate to contact us directly. Thank you again for the opportunity to consider this work for The EMBO Journal, and I look forward to receiving your final version!

With kind regards,

Hartmut

- size of the scale bars that are mandatory for all micrograph panels
- the statistical test used to generate error bars and P-values
- the type error bars (e.g., S.E.M., S.D.)
- the number (n) and nature (biological or technical replicate) of independent experiments underlying each data point
- Figures may not include error bars for experiments with $n < 3$; scatter plots showing individual data points should be used

instead.

9) To facilitate reproducibility and cross-laboratory adoption of methodologies, please structure the Materials & Methods section as outlined in our guide to authors, including a completed Reagents and Tools Table that can be downloaded from our author guidelines as well (<https://www.embopress.org/page/journal/14602075/authorguide#structuredmethods>).

10) Digital image enhancement is acceptable practice, as long as it accurately represents the original data and conforms to community standards. If a figure has been subjected to significant electronic manipulation, this must be clearly noted in the figure legend and/or the 'Materials and Methods' section. The editors reserve the right to request original versions of figures and the original images that were used to assemble the figure. Finally, we generally encourage uploading of numerical as well as gel/blot image source data; for details see: embopress.org/page/journal/14602075/authorguide#sourcedata

At EMBO Press, we ask authors to provide source data for the main manuscript figures. Our source data coordinator will contact you to discuss which figure panels we would need source data for and will also provide you with helpful tips on how to upload and organize the files.

Further information is available in our Guide For Authors:

In the interest of ensuring the conceptual advance provided by the work, we recommend submitting a revision within 3 months (20th Jan 2025). Please discuss the revision progress ahead of this time with the editor if you require more time to complete the revisions. Use the link below to submit your revision:

Link Not Available

Referee #1:

The authors have done an exemplary job in addressing the reviewers' comments and implementing thoughtful revisions to their manuscript. Their responses to all concerns are detailed and well-reasoned, showcasing a deep understanding of the subject matter and a commitment to producing high-quality research. The additional discussion and analyses further strengthen the findings and provide valuable insights. Overall, this manuscript offers a comprehensive and important contribution to the field, and I am confident it will be of great interest and utility to the scientific community. I strongly support its publication.

Referee #3:

Philip et al have addressed most of our previous comments and we appreciate that they conducted additional experiments to

demonstrate the localization efficiency of qTAG system using fluorescent cassettes. They also performed junction PCR and amplicon sequencing to characterize the allelic editing frequency, clonal allelic outcomes, as well as the editing outcomes. However, it would be interesting to see the frequency of off-target outcomes. Overall, the manuscript is much improved, making it a valuable resource paper for researchers seeking practical guidance in CRISPR-mediated endogenous tagging.

Rev_Com_number: RC-2024-02519

New_manu_number: EMBOJ-2024-119161-T

Corr_author: Pelletier

Title: An adaptable plasmid scaffold for CRISPR-based endogenous tagging

GENERAL:

- Please download and complete our author checklist (link provided below).

Response: This has been uploaded in the manuscript submission system.

- Please provide suggestions for a short 'blurb' text prefacing and summing up the conceptual aspect of the study in two sentences (max. 250 characters), followed by 3-5 one-sentence 'bullet points' with brief factual statements of key results of the paper; they will form the basis of an editor-written 'Synopsis' accompanying the online version of the article. Please also upload a synopsis image, which can be used as a "visual title" for the synopsis section of your paper (maybe based on a slimmed-down version of Figure 1?). The image should be in PNG or JPG format, and please make sure that it remains in the modest dimensions of (exactly) 550 pixels wide and 300-600 pixels high.

Response: Please use the following text.

The qTAG system provides a flexible, efficient framework for CRISPR-mediated endogenous tagging, enabling precise protein localization, dynamics, and function studies. Its modular design simplifies tag insertion and accelerates post-editing selection.

- Optimized repair cassettes enable quick isolation of tagged cells after CRISPR editing.
- Versatile cassette designs support both N- and C-terminal tagging.
- Easy modification of tags/selectable markers through built-in restriction sites.
- Demonstrated utility for fluorescence imaging, protein degradation, and epitope tagging.
- Collection available via Addgene, including ready-to-use and customizable constructs.

We have added this on the first page of the revised manuscript with the synopsis image. We have also uploaded the synopsis image at full resolution on the manuscript submission system.

- You shall also receive a separate message from our Source Data curation team, with instructions on how to prepare and upload relevant image and numerical raw data.

Response: This has been done.

TEXT:

- Please upload the manuscript text as an editable DOCX file, without tracking of earlier changes.

Response: The manuscript has been uploaded as an editable DOCX file.

- Please adjust the order of the manuscript sections: Title page with complete author information, Abstract, Keywords, Introduction, Results, Discussion, Methods, Data Availability, Acknowledgements,

Disclosure and Competing Interests Statement, References, Main Figure Legends, Tables, Expanded Figure Legends.

Response: We need to add a Data availability section in the main manuscript

- Please remove the Abbreviations glossary on the front page, and instead make sure to define each abbreviation upon first use of the respective term.

Response: We will remove the abbreviation table from the front page and instead define each abbreviation when it's first mentioned.

-

Please provide a more concise abstract, keeping our regular 175 word length in mind. Also, consider making the title somewhat more explicit, e.g. by mentioning the name of your new system already at that point.

Response: We propose the following new title and shortened abstract. The revised manuscript has been modified accordingly.

Title:

qTAG: An adaptable plasmid scaffold for CRISPR-based endogenous tagging

Abstract:

Endogenous tagging enables the study of proteins within their native regulatory context, typically using CRISPR to insert tag sequences directly into the gene sequence. Here, we introduce qTAG, a collection of repair cassettes that makes endogenous tagging more accessible. The cassettes support N- and C-terminal tagging with commonly used selectable markers and feature restriction sites for easy modification. Lox sites also enable the removal of the marker gene after successful integration. We demonstrate the utility of qTAG with a range of diverse tags for applications in fluorescence imaging, proximity labeling, epitope tagging, and targeted protein degradation. The system includes novel tags like mStayGold, offering enhanced brightness and photostability for live-cell imaging of native protein dynamics. Additionally, we explore alternative cassette designs for conditional expression tagging, selectable knockout tagging, and safe-harbor expression. The plasmid collection is available through Addgene, featuring ready-to-use constructs for common subcellular markers and tagging cassettes to target genes of interest. The qTAG system will serve as an open resource for researchers to adapt and tailor their own experiments.

- Figure Legends: Please make sure to define the error bars in the legends of figures 2d; 5b; 6e, l; and to appropriately label the axis gaps in figure 2d.

Response: We have made the necessary corrections to the above-mentioned figure legends.

- Please note that Materials and Methods need to be described in the main text using our 'Structured Methods' format (for detail, see

<https://www.embopress.org/page/journal/14693178/authorguide#structuredmethods>). The in-text

"Methods" section should contain method and protocol descriptions (ideally using a step-by-step protocol format to facilitate adoption of the methodologies across labs), while all key reagents, experimental models, software and relevant equipment - including their sources and relevant identifiers - should be listed in a separately uploaded Reagents and Tools Table, a template for which can be downloaded from the above section of our Author Guidelines.

Response: The key reagents have been reported in the Reagents and Tools Table template. We believe the methods suffice for the paper. However, if there are points of confusion that you come across, please let us know and we can adapt a point-by-point protocol where needed.

- As we are switching from a free-text author contribution statement towards a more formal statement based on Contributor Role Taxonomy (CRediT) terms, please remove the present Author Contribution section and instead specify each author's contribution(s) directly in the Author Information page of our submission system during upload of the final manuscript. See <https://casrai.org/credit/> for more information.

Response: We have done this and removed the author contribution section in the manuscript.

- Please rename the Competing Interest section into "Disclosure and Competing Interests Statement", in accordance with our updated Guide to Authors (<https://www.embopress.org/competing-interests>)

Response: We have modified the manuscript accordingly.

- Please adjust the format of the reference list and of the in-text citations according to EMBO Journal format (alphabetical order, author name et al + year, first up to 10 authors should be listed, followed by 'et al' ...).

Response: We changed the reference format to EMBO J.

- Please include a dedicated "Data Availability" section at the end of the Material and Methods (suggested wording: "The [structural coordinates | microarray | mass spectrometry] data from this publication have been deposited to the [name of the database] database [URL] and assigned the identifier [accession | permalink | hashtag]."); should there no data deposition to public repositories linked to the study, this should still be stated as "This study includes no data deposited in external repositories."

Response: We have included a statement indication that there is not data deposited in any repository.

- Please make sure to include all relevant funding information not only in the manuscript text, but also in our submission system.

Response : This has been added.

DATA:

- Please refer to our author guide (www.embopress.org/page/journal/14602075/authorguide#expandedview) regarding "supplementary

information", and consider re-organizing the current figures and supplemental figures. We are not limited to 6 main figures, and in addition, we can have up to 5 "Expanded View" figures, whose legends would also need to be in the main text, and which would be type-set and directly visible (expandable) with the HTML version of the paper. My suggestion would be to convert 5 of the current supplementary figures into Expanded View figures (naming/in-text callouts: Figure EV1-5), and to promote some other "supplementary" content into additional main figures/panels.

Response: We will incorporate the five supplementary figures into the main manuscript (Expanded View), while the remaining supplementary figures will be included in the supplementary section

- All main and EV figures should be uploaded as individual files with sufficient resolution/quality for production.

Response: We will upload the individual files in sufficient resolution for the production

- Please double-check that all microscopy panels have had image data inserted, even in the seemingly blank control panels.

Response: We will verify all the microscopy panel for the blank control

- Please convert the four tables into Expanded View Tables (name/callout: Table EV1-4), moving them out of the text and into individual DOCX or XLSX files including their headers/legends.

Response: We will add these tables to the expanded view. (We need to name these as EV1-4)

- Please convert the "Supplemental Data" into Expanded View Datasets (callout: Dataset EV1-2). Dataset EV1 can be uploaded as ZIP archive with its legend included as readme/legend text file; and Dataset EV2 should be uploaded as XLSX file, with its legend & header included in a separate "legend" tab.

Response: We will include the supplementary data in the expanded view, referring to them as "Expanded View data" for the supplementary figures, labeled as EV1-5.

Should you need additional guidance/feedback regarding this final adjustments, please do not hesitate to contact us directly. Thank you again for the opportunity to consider this work for The EMBO Journal, and I look forward to receiving your final version!

Response: We will add Data Availability section prior to submission

- size of the scale bars that are mandatory for all micrograph panels

- the statistical test used to generate error bars and P-values
- the type error bars (e.g., S.E.M., S.D.)
- the number (n) and nature (biological or technical replicate) of independent experiments underlying each data point
- Figures may not include error bars for experiments with $n < 3$; scatter plots showing individual data points should be used instead.

Response: We will add SD, SEM for all the figures where we did not add the SD or SEM

Response: As mentioned earlier we will move five supplementary figures into the expanded view

4) Each main and each Expanded View (EV) figure should be uploaded as individual production-quality files (preferably in .eps, .tif, .jpg formats). For suggestions on figure preparation/layout, please refer to our Figure Preparation Guidelines:

Response: We will upload each main and expanded view figure as an individual file

Response: We will add the response letter as an editable text file in the referee comments section letter

6) Please complete our Author Checklist, and make sure that information entered into the checklist is also reflected in the manuscript; the checklist will be available to readers as part of the Review Process File. A download link is found at the top of our Guide to Authors:

embopress.org/page/journal/14602075/authorguide

Response: We will follow all the guidelines as mentioned on the EMBO website to prepare figures and layout of the manuscript

Response: We will add the ORCID identifier for all the authors listed as (co-)corresponding

Response: As mentioned earlier we will add the supplementary figure to the expanded view

9) To facilitate reproducibility and cross-laboratory adoption of methodologies, please structure

the Materials & Methods section as outlined in our guide to authors, including a completed Reagents and Tools Table that can be downloaded from our author guidelines as well (<https://www.embopress.org/page/journal/14602075/authorguide#structuredmethods>).

Response: We will adopt the material and method sections as outlined in EMBO guideline

10) Digital image enhancement is acceptable practice, as long as it accurately represents the original data and conforms to community standards. If a figure has been subjected to significant electronic manipulation, this must be clearly noted in the figure legend and/or the 'Materials and Methods' section. The editors reserve the right to request original versions of figures and the original images that were used to assemble the figure. Finally, we generally encourage uploading of numerical as well as gel/blot image source data; for details see:

[embopress.org/page/journal/14602075/authorguide#sourcedata](https://www.embopress.org/page/journal/14602075/authorguide#sourcedata)

Response: This is a general point. We will mention it in the figure legend if we have used the any image enhancement

At EMBO Press, we ask authors to provide source data for the main manuscript figures. Our source data coordinator will contact you to discuss which figure panels we would need source data for and will also provide you with helpful tips on how to upload and organize the files.

Further information is available in our Guide For Authors:

In the interest of ensuring the conceptual advance provided by the work, we recommend submitting a revision within 3 months (20th Jan 2025). Please discuss the revision progress ahead of this time with the editor if you require more time to complete the revisions. Use the link below to submit your revision:

<https://emboj.msubmit.net/cgi-bin/main.plex?el=A2Ii3BDuW5A6oFD31A9ftdTYmOrHu8Lo6KBG6OYXf3QY>

Referee #1:

The authors have done an exemplary job in addressing the reviewers' comments and implementing thoughtful revisions to their manuscript. Their responses to all concerns are detailed and well-reasoned, showcasing a deep understanding of the subject matter and a commitment to producing high-quality research. The additional discussion and analyses further strengthen the findings and provide valuable insights. Overall, this manuscript offers a comprehensive and important contribution to the field, and I am confident it will be of great interest and utility to the scientific community. I strongly support its publication.

Response: We sincerely appreciate the reviewer's thoughtful and positive feedback. Your kind words and recognition of our efforts to address the comments and improve the manuscript are greatly valued. Your insights and suggestions have been instrumental in enhancing the quality and clarity of our work. We are grateful for your support and belief in the importance of this research, and we look forward to contributing to the scientific community. Thank you for your time, effort, and encouraging endorsement of our study for publication.

Referee #3:

Philip et al have addressed most of our previous comments and we appreciate that they conducted additional experiments to demonstrate the localization efficiency of qTAG system using fluorescent cassettes. They also performed junction PCR and amplicon sequencing to characterize the allelic editing frequency, clonal allelic outcomes, as well as the editing outcomes. However, it would be interesting to see the frequency of off-target outcomes. Overall, the manuscript is much improved, making it a valuable resource paper for researchers seeking practical guidance in CRISPR-mediated endogenous tagging.

Response: We appreciate that this reviewer feels that we addressed most of their comments to their satisfaction. We agree that it would be of potential interest to evaluate the frequency of off-target outcome, but since this most probably not a qTAG specific issue, and that it is already well known that off-targets are highly gRNA and gene specific, a dedicated and systematic study addressing this question, which fall very much outside the scope of this resource paper, would be necessary.

Dear Dr. Pelletier,

I am pleased to inform you that your manuscript has been accepted for publication in the EMBO Journal.

Yours sincerely,

Rev_Com_number: RC-2024-02519

New_manu_number: EMBOJ-2024-119161R

Corr_author: Pelletier

Title: qTAG: An adaptable plasmid scaffold for CRISPR-based endogenous tagging